# Learning from Stochastically Revealed Preference

**John R. Birge**
The University of Chicago Booth School of Business
John.Birge@chicagobooth.edu

**Xiaocheng Li**
Imperial College Business School, Imperial College London
xiaocheng.li@imperial.ac.uk

**Chunlin Sun**
Institute for Computational and Mathematical Engineering, Stanford University
chunlin@stanford.edu

## Abstract

We study the learning problem of revealed preference in a stochastic setting: a learner observes the utility-maximizing actions of a set of agents whose utility follows some unknown distribution, and the learner aims to infer the distribution through the observations of actions. The problem can be viewed as a single-constraint special case of the inverse linear optimization problem. Existing works all assume that all the agents share one common utility which can easily be violated under practical contexts. In this paper, we consider two settings for the underlying utility distribution: a Gaussian setting where the customer utility follows the von Mises-Fisher distribution, and a $\delta$-corruption setting where the customer utility distribution concentrates on one fixed vector with high probability and is arbitrarily corrupted otherwise. We devise Bayesian approaches for parameter estimation and develop theoretical guarantees for the recovery of the true parameter. We illustrate the algorithm performance through numerical experiments.

## 1 Introduction

The problem of learning from revealed preference refers to the learning of a common utility function for a set of agents based on the observations of the utility-maximizing actions from the agents. The revealed preference problem has a long history in economics [Sam48, Afr67] (See [Var06] for a review). A line of works [BV06, ZR12, BDM$^+$14, ACD$^+$15] formulate the problem as a learning problem with two objectives: (i) rationalizing a set of observations, i.e., to find a utility function which explains a set of past observations; (ii) predicting the future behavior of a utility-maximization agent. Mathematically, the action of the agents is modeled by an optimization problem that maximizes a linear (or concave) utility function subject to one linear budget constraint. The learner (decision maker) aims to learn the unknown utility function through a set of observations of the constraints and the optimal solutions. The problem can be viewed as a single-constraint special case of the inverse optimization problem [AO01] which covers a wider range of applications: geoscience [BT92], finance [BGP12], market analysis [BHP17], energy [ASS18], etc.

In this paper, we study the problem under a stochastic setting where the agents have a linear utility function randomly distributed according to some unknown distribution. Such a stochastic setting is well-motivated by some application context where the agents are customers and the constraint models the prices and the customer's budget. The optimal solution encodes the customer's purchase

36th Conference on Neural Information Processing Systems (NeurIPS 2022).

behavior and the stochastic utility (objective function) captures the heterogeneity of the customer preference for the products. The goal of learning in this stochastic setting thus becomes to learn the utility distribution through observations of the actions. To the best of our knowledge, we provide the first result of learning a stochastic utility for the revealed preference problem and even in the more general literature of the inverse optimization problem.

**Related Literature:** The existing approaches to the problem can be roughly divided into two categories.

Query-based: In a query-based model, the learner aims to learn the utility function by querying an oracle for the agent's optimal actions, and the goal is to derive the sample complexity guarantee for a sufficiently accurate estimation of the utility function. [BV06] initiates this line of research and studies a statistical setup where the input data is a set of observations and the learner's performance is evaluated by sample complexity bounds. [ZR12] studies the case of a linear or linearly separable concave utility function, and [BDM+14] generalizes the setting and devises learning algorithms for several classes of utility functions. Other than a statistical setup where the observations are sampled from some distribution, both of these two works study an "active" learning setting where the learner has the power to choose the linear constraint (set the prices of the products). Some subsequent works along this line study the associated revenue management problem [ACD+15] and a game-theoretic setting [DRS+18] where the agents act strategically to hide the true actions.

Optimization-based: The optimization-based approach is usually adopted in the literature of inverse optimization, and some algorithms developed therein can be applied to the special case of the revealed preference problem. [ZL96] and [AO01] study the inverse optimization with one single observation and develop linear programming formulations to solve the problem. Later, [KWB11] and [ASS18] study the statistical (or data-driven) setting where the observations are sampled from some distribution. Specifically, [ASS18] considers a setting where the optimal actions of the agents are contaminated with some independent noises, but all the agents still follow a common utility parameter vector. [MESAHK18] studies the distributional robust version of the problem and [BFL21] considers a contextual formulation. A recent line of works [BMPS18, DCZ18, DZ20, CKK20] cast the inverse optimization problem in an online context and develop (online) gradient-based algorithms.

As we understand, all the existing algorithms and analyses under this topic rely on the assumption that all the agents share one common utility function (or a common utility parameter vector), and thus can fail in the stochastic setting. In this paper, we formulate the problem in Section 2 and focus on the statistical data input where the budget constraint is sampled from some unknown distribution. We consider two stochastic settings: a Gaussian setting in Section 3 and a $\delta$-corruption setting in Section 4. We conclude with numerical experiments and discuss (i) how the results can be generalized to the inverse optimization problem and (ii) the implications for the query-based model where the learner has the power to choose the budget constraints.

## 2 Model Setup

Consider a customer who purchases a bundle of products subject to some budget constraint. The customer's utility-maximizing action can be modeled by the following linear program:

$$\mathrm{LP}(\boldsymbol{u}, \boldsymbol{a}, b) := \max_{\boldsymbol{x}} \ \sum_{i=1}^{n} u_i x_i$$

$$\text{s.t.} \ \sum_{i=1}^{n} a_i x_i \leq b, \quad 0 \leq x_i \leq 1, i = 1, ..., n,$$

where $\boldsymbol{u} = (u_1, ..., u_n) \in \mathbb{R}^n$, $\boldsymbol{a} = (a_1, ..., a_n) \in \mathbb{R}_+^n$, and $b \in \mathbb{R}_+$ are the inputs of the LP. Here the decision variables $\boldsymbol{x}$ encode the purchase decisions where a partial purchase is allowed, $u_i$ denotes the customer's utility for the $i$-th product, and $a_i$ denotes the price or cost of purchasing the $i$-th product. The right-hand-side $b$ denotes the budget of the customer.

Throughout this paper, we make the following assumption.

**Assumption 1.** *We assume:*

- *The utility vector $\boldsymbol{u}$ follows some unknown distribution $\mathcal{P}_u$.*

- *The LP's input $(\boldsymbol{a}, b)$ follows some unknown distribution $\mathcal{P}_{\boldsymbol{a},b}$ independent of $\mathcal{P}_u$.*

- *There exists $\underline{a} > 0$ such that $a_i \in [\underline{a}, 1]$ almost surely for $i = 1, ..., n$.*

Our goal is to infer the distribution through observations of customers' optimal actions. Mathematically, we aim to estimate the distribution of $\mathcal{P}_u$ through the dataset

$$\mathcal{D}_T = \{(\boldsymbol{x}_t^*, \boldsymbol{a}_t, b_t)\}_{t=1}^T .$$

Here the $t$-th sample corresponds to an unobservable $\boldsymbol{u}_t$ generated from $\mathcal{P}_u$ and $\boldsymbol{x}_t^*$ is one optimal solution of $\mathrm{LP}(\boldsymbol{u}_t, \boldsymbol{a}_t, b_t)$. Due to the scale invariance of the utility vector, we restrict the distribution $\mathcal{P}_u$ to the unit sphere $\mathcal{S}^{n-1} = \{\boldsymbol{u} : \|\boldsymbol{u}\|_2 = 1\}$. In the following two sections, we consider two settings: (i) Gaussian: $\mathcal{P}_u$ follows the von Mises–Fisher distribution, i.e., the restriction of a multivariate Gaussian distribution to the unit sphere; (ii) $\delta$-corruption: $\mathcal{P}_u$ concentrates on one point $\boldsymbol{u}^*$ with probability $1 - \delta$ and follows an arbitrarily corrupted distribution with probability $\delta$.

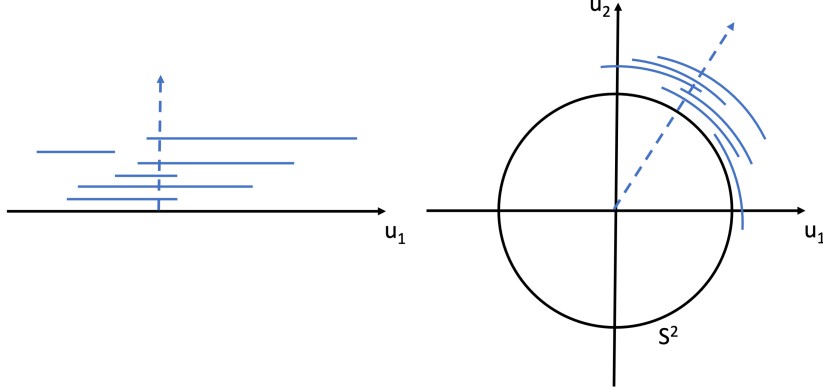

Figure 1: Visualizing the challenge of the problem in 1-D and 2-D.

**The challenge of the problem.** Each observation $(\boldsymbol{x}_t^*, \boldsymbol{a}_t, b_t)$ prescribes a region $\mathcal{U}_t \subset \mathcal{S}^{n-1}$,

$$\mathcal{U}_t := \left\{\boldsymbol{u} \in \mathcal{S}^{n-1} : \boldsymbol{x}_t^* \text{ is an optimal solution of } \mathrm{LP}(\boldsymbol{u}, \boldsymbol{a}_t, b_t)\right\} .$$

The set $\mathcal{U}_t$ captures all the possible values of $\boldsymbol{u}_t$ that is consistent with the $t$-th observation. The following lemma states that the set $\mathcal{U}_t$ can be expressed by a group of linear constraints.

**Lemma 1.** *For each $\mathcal{U}_t$, there exists a matrix $\boldsymbol{V}_t$ and a vector $\boldsymbol{w}_t$ such that*

$$\mathcal{U}_t = \left\{\boldsymbol{u} \in \mathcal{S}^{n-1} : \boldsymbol{V}_t \boldsymbol{u} \le \boldsymbol{w}_t\right\} .$$

In the deterministic setting of the revealed preference problem, all the $\boldsymbol{u}_t$'s are identical and the learning problem is thus reduced to finding one feasible $\boldsymbol{u}$ in the set of $\cap_{t=1}^T \mathcal{U}_t$. But in a stochastic setting, it may happen that the set of $\cap_{t=1}^T \mathcal{U}_t$ is empty. Figure 1 provides a conceptual visualization of this challenge of "empty intersection". Each blue solid segment denotes one such $\mathcal{U}_t$ and the blue dashed line represents a value of $\boldsymbol{u}$ that appears most frequently in these $\mathcal{U}_t$'s. We remark that the figure is just for illustrative purpose as the problem may not be well-defined in the 1-dimensional case.

From an estimation viewpoint, the goal is to estimate the distribution of $\mathcal{P}_u$ without the knowledge of the realized samples $\boldsymbol{u}_t$'s, but merely with the knowledge of $\mathcal{U}_t$ to which $\boldsymbol{u}_t$ belongs. The sample efficiency of the estimation procedure is naturally contingent on the dispersion of $\mathcal{U}_t$ which is essentially determined by the generation of $(\boldsymbol{a}_t, b_t)$. For example, if all the $\mathcal{U}_t$'s coincide with each other, then one can hardly learn much about the underlying $\mathcal{P}_u$. In this paper, we aim to pinpoint conditions for $\mathcal{P}_{\boldsymbol{a},b}$ such that the learning of $\mathcal{P}_u$ is possible. Also, an alternative way to measure the estimation accuracy is to evaluate the predictive performance of the estimated model on new observations generated from $\mathcal{P}_{\boldsymbol{a},b}$, and such performance bounds generally bear less dependency on the distribution of $\mathcal{P}_{\boldsymbol{a},b}$. We also provide theoretical guarantees in this sense.

# 3 Gaussian Setting

In this section, we consider a setting where the distribution $\mathcal{P}_u$ follows the von Mises-Fisher distribution parameterized by $\boldsymbol{\theta} = (\boldsymbol{\mu}, \kappa)$ with the density function

$$f(\boldsymbol{u}; \boldsymbol{\theta}) := \frac{\exp\left(-\kappa \boldsymbol{\mu}^\top \boldsymbol{u}\right)}{\int_{\boldsymbol{u} \in \mathcal{S}^{n-1}} \exp(-\kappa \boldsymbol{\mu}^\top \boldsymbol{u}) d\boldsymbol{u}} \propto \exp\left(-\kappa \boldsymbol{\mu}^\top \boldsymbol{u}\right).$$

Here the vector $\boldsymbol{\mu} \in \mathbb{R}^n$ represents the mean direction and the parameter $\kappa > 0$ controls the concentration of the distribution. The deterministic setting of the revealed preference problem can be viewed as the case when $\kappa = \infty$ and then the distribution degenerates to a point-mass distribution on the unit sphere. Denote the true parameters of the distribution $\mathcal{P}_u$ by $\boldsymbol{\theta}^* = (\boldsymbol{\mu}^*, \kappa^*)$. Then the likelihood of the dataset $\mathcal{D}_T$ under a parameter $\boldsymbol{\theta}$ is

$$\mathbb{P}(\mathcal{D}_T | \boldsymbol{\theta}) := \prod_{t=1}^T \mathbb{P}\left((\boldsymbol{x}_t^*, \boldsymbol{a}_t, b_t) | \boldsymbol{\theta}\right) = \prod_{t=1}^T \int_{\boldsymbol{u} \in \mathcal{U}_t} f(\boldsymbol{u}; \boldsymbol{\theta}) du.$$

We remark that the maximum likelihood approach cannot be applied here for two reasons. First, the integration of $f(\boldsymbol{u}; \boldsymbol{\theta})$ over the region $\mathcal{U}_t$ is not closed-form. The first point is not only pertaining to the Gaussian parameterization of $\mathcal{P}_u$. The scale-invariant property of the utility vector restricts the distribution $\mathcal{P}_u$ to a unit sphere or a simplex, and consequently, the likelihood function inevitably involves the non-closed-form integration. This issue can be partially resolved by using the Monte Carlo method to approximate the integration, and a good thing is that the same integrand is shared across all the observations. Second, the likelihood function is not analytical in $\boldsymbol{\theta}$. Thus this prevents the usage of gradient-based algorithms to solve the problem and also makes it difficult to derive theoretical guarantees for the maximum likelihood estimator.

We propose a Bayesian perspective for the problem: instead of identifying the parameter that maximizes the likelihood function, we directly draw samples from the posterior distribution. We will see shortly that the approach can be justified through a concentration property of the posterior distribution. Suppose we have a prior distribution $\mathbb{P}_0(\boldsymbol{\theta})$ and then we can define the posterior distribution by

$$\mathbb{P}_T(\boldsymbol{\theta}) := \frac{\mathbb{P}_0(\boldsymbol{\theta}) \cdot \mathbb{P}(\mathcal{D}_T | \boldsymbol{\theta})}{\mathbb{P}(\mathcal{D}_T)}$$

$$\propto \mathbb{P}_0(\boldsymbol{\theta}) \cdot \prod_{t=1}^T \int_{\boldsymbol{u} \in \mathcal{U}_t} f(\boldsymbol{u}; \boldsymbol{\theta}) du.$$

With slight abuse of notation, we use $\mathbb{P}_T(\cdot)$ (or $\mathbb{P}_0(\cdot)$) to refer to both the density function and the probability measure of the posterior (or prior) distribution. We make the following assumption on the prior distribution.

**Assumption 2.** *We assume the concentration parameter $\kappa^* \in (\underline{\kappa}, \bar{\kappa})$ where $\underline{\kappa}, \bar{\kappa}$ are two known positive constants. The prior distribution $\mathbb{P}_0(\boldsymbol{\theta})$ is a uniform distribution on $\mathcal{S}^{n-1} \times (\underline{\kappa}, \bar{\kappa})$.*

**Theorem 1.** *Let*

$$\Theta_T := \left\{ \boldsymbol{\theta} \in \mathcal{S}^{n-1} \times (\underline{\kappa}, \bar{\kappa}) : \mathcal{W}\left(\mathbb{P}\left((\boldsymbol{x}_t^*, \boldsymbol{a}_t, b_t) | \boldsymbol{\theta}\right), \mathbb{P}\left((\boldsymbol{x}_t^*, \boldsymbol{a}_t, b_t) | \boldsymbol{\theta}^*\right)\right) \leq \max\left(8, 8\bar{\kappa}\right) \frac{n \cdot \log T}{T^{1/2}} \right\}$$

*where $\mathcal{W}(\cdot, \cdot)$ is the Wasserstein distance between two distributions supported on $\mathcal{X} \times \mathbb{R}_+^n \times \mathbb{R}_+$ equipped with Euclidean metric. Then, under Assumptions 1-2,*

$$1 - \mathbb{P}_T(\Theta_T) \to 0 \text{ in probability as } T \to \infty.$$

*Specifically, the following inequality holds*

$$\mathbb{E}\left[\mathbb{P}_T(\Theta_T)\right] \geq 1 - \frac{3}{T}.$$

*where the expectation is taken with respect to the random distribution $\mathbb{P}_T(\cdot)$ (essentially, with respect to the dataset $\mathcal{D}_T$.)*

Theorem 1 justifies the approach of posterior sampling. We first remark that the Bayesian sampling approach is just proposed to estimate the parameters, but all the theoretical results are stated in frequentist language. The proof of Theorem 1 follows the standard analysis of the convergence of the posterior distribution [GGVDV00, CDBW21]. While similar results should also hold for other underlying distribution of $\mathcal{P}_u$, the von Mises-Fisher distribution provides much analytical convenience in deriving the bound. Each $\boldsymbol{\theta}$, together with the distribution of $\mathcal{P}_{\boldsymbol{a},b}$, defines a distribution over the space of $(\boldsymbol{x}_t^*, \boldsymbol{a}_t, b_t)$. As we use observations $(\boldsymbol{x}_t^*, \boldsymbol{a}_t, b_t)$'s to identify the true $\boldsymbol{\theta}^*$, the set $\Theta_T$ defines a set of indistinguishable $\boldsymbol{\theta}$'s based on the Wasserstein distance between distributions of $(\boldsymbol{x}_t^*, \boldsymbol{a}_t, b_t)$. The set $\Theta_T$ shrinks as $T \to \infty$. The posterior sampling approach samples from the distribution $\mathbb{P}_T(\cdot)$, and Theorem 1 states that the samples will be concentrated in set $\Theta_T$ with high probability. The posterior distribution $\mathbb{P}_T(\cdot)$ is dependent on the dataset $\mathcal{D}_T$, so it is a random distribution itself and the results in Theorem 1 are stated in either convergence in probability or expectation. As a side note, the Wasserstein distance in the theorem is not critical and it can be replaced with other distances such as the total variation distance and the Hellinger distance.

Intuitively, Theorem 1 says that for some $\boldsymbol{\theta}$ such that the likelihood distribution $\mathbb{P}\left((\boldsymbol{x}_t^*, \boldsymbol{a}_t, b_t)|\boldsymbol{\theta}\right)$ differs from $\mathbb{P}\left((\boldsymbol{x}_t^*, \boldsymbol{a}_t, b_t)|\boldsymbol{\theta}^*\right)$ to a certain extent, the posterior $\mathbb{P}_T(\cdot)$ is unlikely to generate such $\boldsymbol{\theta}$. In other words, the posterior distribution identifies the true $\boldsymbol{\theta}^*$ up to some "equivalence" in the likelihood distribution space. The following corollary formalizes this intuition that if there is an equivalence between the likelihood distribution space and the underlying parameter space, then the posterior distribution is capable of identifying the true parameter.

**Corollary 1.** *Suppose*

$$\mathcal{W}\left(\mathbb{P}\left((\boldsymbol{x}_t^*, \boldsymbol{a}_t, b_t)|\boldsymbol{\theta}\right), \mathbb{P}\left((\boldsymbol{x}_t^*, \boldsymbol{a}_t, b_t)|\boldsymbol{\theta}^*\right)\right) > 0$$

*for all $\boldsymbol{\theta} \neq \boldsymbol{\theta}^* \in \mathcal{S}^{n-1} \times [\underline{\kappa}, \bar{\kappa}]$. Then the posterior distribution $\mathbb{P}_T(\cdot)$ will converge to the point-mass distribution on $\boldsymbol{\theta}^*$ almost surely as $T \to \infty$. Moreover, suppose there exists a constant $L > 0$ satisfying*

$$\mathcal{W}\left(\mathbb{P}\left((\boldsymbol{x}_t^*, \boldsymbol{a}_t, b_t)|\boldsymbol{\theta}\right), \mathbb{P}\left((\boldsymbol{x}_t^*, \boldsymbol{a}_t, b_t)|\boldsymbol{\theta}^*\right)\right) \geq L \cdot \|\boldsymbol{\theta} - \boldsymbol{\theta}^*\|_2, \tag{1}$$

*for all $\boldsymbol{\theta} \neq \boldsymbol{\theta}^* \in \mathcal{S}^{n-1} \times [\underline{\kappa}, \bar{\kappa}]$.*

*Under Assumptions 1-2, the following inequality holds with probability no less than $1 - \frac{6L}{n \cdot T^{1/2} \log T}$,*

$$\mathbb{E}_T\left[\|\boldsymbol{\theta}_T - \boldsymbol{\theta}^*\|_2\right] \leq \max\left(9, 9\bar{\kappa}\right) \frac{n \cdot \log T}{L \cdot T^{1/2}}$$

*where $\boldsymbol{\theta}_T$ is sampled from the posterior distribution $\mathbb{P}_T(\cdot)$.*

The corollary states that when there is some equivalence between the likelihood distribution space and the parameter space as (1), the true parameter is identifiable. The first part of the corollary states a consistency result that as long as all the $\boldsymbol{\theta} \neq \boldsymbol{\theta}^*$ are distinguishable from $\boldsymbol{\theta}^*$ through the likelihood function, then the posterior sampling will eventually identify the true $\boldsymbol{\theta}^*$. The second part relates to the convergence rate with an equivalence parameter $L$.

In Assumption 1, we assume the constraint input $(\boldsymbol{a}_t, b_t)$ is generated from some distribution $\mathcal{P}_{\boldsymbol{a},b}$. We note that Theorem 1 and Corollary 1 hold without any additional assumption on $\mathcal{P}_{\boldsymbol{a},b}$, but the space topology of the likelihood distribution is highly dependent on $\mathcal{P}_{\boldsymbol{a},b}$. Specifically, a different distribution of $(\boldsymbol{a}_t, b_t)$ determines the separateness of the parameter space through affecting the value of $L$ in (1) or even its existence. The value $L$ of a specific distribution of $\mathcal{P}_{\boldsymbol{a},b}$ can be examined through simulation. So if the learner has some flexibility in choosing the distribution of $\mathcal{P}_{\boldsymbol{a},b}$, the optimal choice would be the one that corresponds to a larger value of $L$. If the constraint input $(\boldsymbol{a}_t, b_t)$ is not randomly generated but can be actively chosen as the query-based preference learning problem, the results in Theorem 1 and Corollary 1 still hold by conditioning on all the $(\boldsymbol{a}_t, b_t)$'s. Unlike the deterministic case where the utility vector $\boldsymbol{u}$ is fixed for all the observations, the stochastic nature of the problem setup here makes it generally very complicated to fully extract the benefit of designing $(\boldsymbol{a}_t, b_t)$'s by the learner. We leave it as a future open question.

**Corollary 2.** *Suppose $\mathcal{P}_{\boldsymbol{a},b}$ is a discrete distribution with a finite support. Let $(\boldsymbol{a}, b)$ be a new sample from $\mathcal{P}_{\boldsymbol{a},b}$, i.e., independent from the dataset $\mathcal{D}_T$, and let $\boldsymbol{\theta}_T = (\boldsymbol{\mu}_T, \kappa_T)$ be a sample from the posterior distribution $\mathbb{P}_T(\cdot)$. Denote $\tilde{\boldsymbol{x}}^*$ and $\boldsymbol{x}^*$ as the optimal solutions of $LP(\boldsymbol{\mu}_T, \boldsymbol{a}, b)$ and $LP(\boldsymbol{\mu}^*, \boldsymbol{a}, b)$, respectively. Then, under Assumptions 1-2, the following inequality holds with probability no less than $1 - \frac{6}{\sqrt{n} \cdot T^{1/2} \log T}$,*

$$\mathbb{E}\left[\|\tilde{\boldsymbol{x}}^* - \boldsymbol{x}^*\|_2\right] \leq \max\left(16, 16\bar{\kappa}\right) \frac{n \cdot \log T}{T^{1/2}},$$

*where the expectation is taken with respect to both the posterior distribution $\mathbb{P}_T(\cdot)$ and $(\boldsymbol{a}, b)$.*

Corollary 2 provides an upper bound on the predictive performance of the posterior distribution. Specifically, we want to predict the optimal solution of a linear program specified by $\boldsymbol{\mu}^*$ (proportionally to $\mathbb{E}[\boldsymbol{u}]$) and a new sample of the constraint $(\boldsymbol{a}, b)$, and the prediction $\tilde{\boldsymbol{x}}^*$ is based on a posterior sample. We know from Theorem 1 that the posterior distribution concentrates on those $\boldsymbol{\theta}$'s that are indistinguishable from the true $\boldsymbol{\theta}^*$ in terms of the likelihood. Speaking of the predictive performance, we only concern the distribution of the optimal solution (equivalently, the likelihood), but do not require the identification of exact true $\boldsymbol{\theta}^*$, so Corollary 2 does not require the condition (1) to hold. Intuitively, the prediction of the optimal solution on a new observation $(\boldsymbol{a}, b)$ can be viewed as a condition distribution of the optimal solution given $(\boldsymbol{a}, b)$. While the definition of $\Theta_T$ in Theorem 1 concerns the joint distribution of the optimal solution and $(\boldsymbol{a}, b)$, the finite-support condition on $\mathcal{P}_{\boldsymbol{a},b}$ in Corollary 2 transforms the result on the joint distribution to the conditional distribution.

## 4 $\delta$-Corruption Case

In this section, we consider a setting where the utility vector is specified by

$$\boldsymbol{u}_t = \begin{cases} \boldsymbol{u}^*, & \text{w.p. } 1 - \delta, \\ \mathcal{P}'_u, & \text{w.p. } \delta, \end{cases} \tag{2}$$

where $\boldsymbol{u}^* \in \mathcal{S}^{n-1}$ is a fixed vector, $\delta \in [0, 1]$, and $\mathcal{P}'_u$ is an arbitrary distribution that corrupts the inference of $\boldsymbol{u}^*$. The deterministic setting of the revealed preference problem in literature can be viewed as the case of $\delta = 0$, and the Gaussian setting in the previous section can be viewed as the case of $\delta = 1$ and $\mathcal{P}'_u$ being the von-Mises Fisher distribution. In this setting, we do not aim to learn the distribution of $\mathcal{P}'_u$, but rather our goal is to identify the vector $\boldsymbol{u}^*$ using the dataset $\mathcal{D}_T$.

A natural idea to estimate $\boldsymbol{u}^*$ is by solving the following optimization problem:

$$\text{OPT}_\delta := \max_{\boldsymbol{u} \in \mathcal{S}^{n-1}} \sum_{t=1}^{T} I_{\mathcal{U}_t}(\boldsymbol{u})$$

where the indicator function $I_{\mathcal{E}}(e) = 1$ if $e \in \mathcal{E}$ and $I_{\mathcal{E}}(e) = 0$ otherwise. The rationale for the optimization problem is that for the $t$-th observation, a vector $\boldsymbol{u}$ is consistent with the observation, i.e., $\boldsymbol{x}_t^*$ is the optimal solution of $\text{LP}(\boldsymbol{u}, \boldsymbol{a}_t, b_t)$, if and only if $I_{\mathcal{U}_t}(\boldsymbol{u}) = 1$. Thus the optimization problem finds a vector $\boldsymbol{u}$ that is consistent with the maximal number of observations. The objective function is discontinuous in $\boldsymbol{u}$, so we propose the simulated annealing algorithm – Algorithm 2 to solve for its optimal solution.

We first build some connection between the optimization problem $\text{OPT}_\delta$ and that of the deterministic setting with $\delta = 0$. Let $\bar{\boldsymbol{x}}_t^*$ be the optimal solution of $\text{LP}(\boldsymbol{u}^*, \boldsymbol{a}_t, b_t)$ and define

$$\bar{\mathcal{U}}_t := \left\{ \boldsymbol{u} \in \mathcal{S}^{n-1} : \bar{\boldsymbol{x}}_t^* \text{ is an optimal solution of } \text{LP}(\boldsymbol{u}, \boldsymbol{a}_t, b_t) \right\}.$$

Then the deterministic setting of the revealed preference problem solves

$$\bar{\text{OPT}}_\delta := \max_{\boldsymbol{u} \in \mathcal{S}^{n-1}} \sum_{t=1}^{T} I_{\bar{\mathcal{U}}_t}(\boldsymbol{u}).$$

By the setup of the problem, $\boldsymbol{u}^*$ is an optimizer of $\bar{\text{OPT}}_\delta$ and the optimal objective value is $T$. The following proposition establishes that the optimization problem $\text{OPT}_\delta$ is a contaminated version of $\bar{\text{OPT}}_\delta$ and the effect that the contamination has on the objective function can be bounded using $\delta$.

**Proposition 1.** *Under Assumption 1, the following inequality holds*

$$\mathbb{P}\left( \max_{\boldsymbol{u} \in \mathcal{S}^{n-1}} \left| \frac{1}{T} \sum_{t=1}^{T} I_{\mathcal{U}_t}(\boldsymbol{u}) - \frac{1}{T} \sum_{t=1}^{T} I_{\bar{\mathcal{U}}_t}(\boldsymbol{u}) \right| \le \delta + \frac{\log T}{\sqrt{T}} \right) \le \frac{1}{T}.$$

When the constraints $(\boldsymbol{a}_t, b_t)$'s are generated from some distribution $\mathcal{P}_{\boldsymbol{a},b}$, it can happen that there exist some vectors $\boldsymbol{u}'$ that are indistinguishable from $\boldsymbol{u}^*$ based on the observations $\mathcal{D}_T$ as in the

previous Gaussian case. So, we do not hope for an exact recovery of $\boldsymbol{u}^*$, but alternatively, we aim to derive a generalization bound for our estimator $\hat{\boldsymbol{u}}$. Specifically, we define and analyze the accuracy

$$\text{Acc}(\hat{\boldsymbol{u}}) := \mathbb{E}\left[I_{\mathcal{U}}(\hat{\boldsymbol{u}})\right] \text{ with } \mathcal{U} := \left\{\boldsymbol{u} \in \mathcal{S}^{n-1} : \boldsymbol{x}^* \text{ is an optimal solution of } \text{LP}(\boldsymbol{u}_{\text{new}}, \boldsymbol{a}, b)\right\}$$

where $\hat{\boldsymbol{u}}$ is our estimator of $\boldsymbol{u}^*$, $(\boldsymbol{a}, b)$ is a new sample from the distribution $\mathcal{P}_{\boldsymbol{a},b}$, $\boldsymbol{u}_{\text{new}}$ is a new sample following the law of (2), and $\boldsymbol{x}^*$ is the optimal solution of $\text{LP}(\boldsymbol{u}, \boldsymbol{a}, b)$. In other words, the quantity captures the probability that $\hat{\boldsymbol{u}}$ is consistent with a new (unseen) observation, and we know that for the true parameter, $\text{Acc}(\boldsymbol{u}^*) \geq 1 - \delta$, which serves as a performance benchmark.

The challenge for deriving a bound on $\text{Acc}(\hat{\boldsymbol{u}})$ arises from the discontinuity of the objective function $\text{OPT}_\delta$. The existing methods for deriving generalization bound largely rely on the continuity and the Lipschitzness of the loss function. To make it worse, from Lemma 1, we know that $\mathcal{U}_t$ is specified by $(\boldsymbol{V}_t, \boldsymbol{w}_t)$ and the $\boldsymbol{V}_t$'s are of different dimensions for different $t$'s. To overcome these challenges, we devise the following $\gamma$-margin objective function. Specifically, we first define a parameterized version of $\mathcal{U}_t$ by

$$\mathcal{U}_t(\gamma) := \left\{\boldsymbol{u} \in \mathcal{S}^{n-1} : \boldsymbol{V}_t \boldsymbol{u} \leq \boldsymbol{w}_t - \gamma \boldsymbol{e}\right\}$$

where $\gamma$ is a positive constant and $\boldsymbol{e}$ is an all-one vector. It is obvious that $\mathcal{U}_t(\gamma) \subset \mathcal{U}_t$. Accordingly, we define the $\gamma$-margin optimization problem by

$$\text{OPT}_\delta(\gamma) := \max_{\boldsymbol{u} \in \mathcal{S}^{n-1}} \sum_{t=1}^{T} I_{\mathcal{U}_t(\gamma)}(\boldsymbol{u}).$$

**Proposition 2.** *Under Assumption 1, the following inequality holds with probability no less than $1 - \epsilon$,*

$$\max_{\boldsymbol{u} \in \mathcal{S}^{n-1}} \frac{1}{T} \sum_{t=1}^{T} I_{\mathcal{U}_t(\gamma)}(\boldsymbol{u}) - \text{Acc}(\boldsymbol{u}) \leq 4\sqrt{\frac{\log(T)}{\underline{a}^2 \gamma^2 T}} + 6\sqrt{\frac{\log(T/\epsilon)}{T}}$$

*for $\epsilon \in (0, 1)$.*

Proposition 2 relates the generalization accuracy of any arbitrary $\boldsymbol{u}$ with the corresponding objective value of the $\gamma$-margin optimization problem. As $\gamma$ increases, the objective function will decrease, so the right-hand-side becomes tighter. Importantly, the accuracy is defined by the original indicator function (or equivalently, $\mathcal{U}_t$), while the objective value is defined by the $\gamma$-margin indicator function (or equivalently, $\mathcal{U}_t(\gamma)$). The implication is that when we optimize the $\gamma$-margin objective, we can still obtain a bound on the original accuracy $\text{Acc}(\boldsymbol{u})$ for sufficiently large $\gamma$.

**Theorem 2.** *Suppose $\mathcal{P}_{\boldsymbol{a},b}$ is a continuous distribution and it has a density function upper bounded by $\bar{p}$. Then, under Assumption 1, the following inequality holds for sufficient large $T$*

$$\mathbb{P}\left(\text{Acc}(\hat{\boldsymbol{u}}) \geq 1 - \delta - \frac{n\bar{p}}{\min_{i:u_i^* \neq 0} |u_i^*| \cdot T^{1/4}} - \frac{20n \log(T)}{\underline{a} T^{1/4}}\right) \geq 1 - \frac{4}{T},$$

*where $\hat{\boldsymbol{u}}$ is one optimal solution of $\text{OPT}_\delta(\gamma)$ with $\gamma = \frac{1}{4nT^{1/4}}$.*

Theorem 2 states a generalization bound on the accuracy of $\hat{\boldsymbol{u}}$ for continuous distributions of $(\boldsymbol{a}, b)$. From Proposition 2, a larger $\gamma$ leads to a smaller gap between the accuracy and the $\gamma$-margin objective function. Meanwhile, a smaller $\gamma$ leads to a smaller gap between the optimal objective value $\text{OPT}_\delta(\gamma)$ and $1 - \delta$. In the extreme case of $\gamma = 0$, $\mathbb{E}[\text{OPT}_\delta(0)] = \text{Acc}(\boldsymbol{u}^*) \geq 1 - \delta$. Theorem 2 optimizes the value of $\gamma$ to trade off these two aspects. We note that the continuous distribution and the upper bound on the density function make it possible to bound the gap of the second aspect. Finally, we remark that the design of $\gamma$-margin loss function is inspired from the max-margin classifier, but the analysis is entirely different. For the max-margin classifier, the introduction of the margin aims to make the underlying loss function 1-Lipschitz so that a generalization bound using Rademacher complexity can be derived. But for here, our $\gamma$-margin objective function is still a discontinuous one.

## 5 Computational Aspects and Discussions

In the previous section, we developed theoretical results for both Gaussian and $\delta$-corruption settings. Now we discuss computational aspects with respect to the sampling of the posterior $\mathbb{P}_T(\cdot)$ and the

---

**Algorithm 1** Posterior Sampling for the Gaussian Setting

---

1: Input: dataset $\mathcal{D}_T = \{(\boldsymbol{x}_t^*, \boldsymbol{a}_t, b_t)\}_{t=1}^T$, number of iterations $K$
2: Initialize $\boldsymbol{\theta}^{(0)}$ by randomly sampling from the prior distribution $\mathbb{P}_0(\boldsymbol{\theta})$
3: **for** $k = 1, ..., K$ **do**
4:     Draw a random $\boldsymbol{\theta}'$ from a pre-determined proposal distribution $\mathbb{Q}(\boldsymbol{\theta}'|\boldsymbol{\theta}^{(k-1)})$
5:     Compute the acceptance rate:

$$r = \min\left\{\frac{\mathbb{P}_T(\boldsymbol{\theta}')}{\mathbb{P}_T(\boldsymbol{\theta}^{(k-1)})}, 1\right\}$$

6:     Set

$$\boldsymbol{\theta}^{(k)} = \begin{cases} \boldsymbol{\theta}', & \text{w.p. } r \\ \boldsymbol{\theta}^{(k-1)}, & \text{w.p. } 1-r \end{cases}$$

7:
8: **end for**
9: Output: $\boldsymbol{\theta}^{(K)}$

---

optimization of $\text{OPT}_\delta(\gamma)$. As mentioned earlier, the posterior sampling removes the complication of optimizing over $\boldsymbol{\theta}$ in the maximum likelihood estimation, but still inevitably needs to deal with the sampling and numeric approximation of the likelihood function. Algorithm 1 describes a standard Metropolis–Hastings algorithm to sample from the posterior distribution $\mathbb{P}_T(\cdot)$. In the numerical experiments, we choose the proposal distribution $\mathbb{Q}$ to be a Gaussian random perturbation, i.e., $\boldsymbol{\theta}' = \text{Proj}(\boldsymbol{\theta}^{(k-1)} + \boldsymbol{\epsilon})$ where $\boldsymbol{\epsilon}$ follows a Gaussian distribution and the projection ensures that $\boldsymbol{\theta}'$ stays on the sphere $\mathcal{S}^{n-1}$. For the acceptance ratio, as the posterior distribution is not in closed form, a Monte Carlo subroutine is needed to estimate the ratio.

---

**Algorithm 2** Simulated annealing algorithm for $\delta$-corruption

---

1: Input: dataset $\mathcal{D}_T = \{(\boldsymbol{x}_t^*, \boldsymbol{a}_t, b_t)\}_{t=1}^T$, margin $\gamma$, number of iterations $K$, interval length $\tau$
2: Initialize an initial (temperature) $\eta > 0$ and the reduction rate $c \in (0, 1)$
3: Randomly generate the first estimate $\boldsymbol{u}^{(0)}$
4: **for** $k = 1, ..., K$ **do**
5:     **if** $k \mod \tau = 0$ **then**
6:         Update $\eta \leftarrow c \cdot \eta$
7:     **end if**
8:     Draw a proposal $\boldsymbol{u}'$ from a predetermined proposal distribution $\mathbb{Q}(\boldsymbol{u}'|\boldsymbol{u}^{(k-1)})$
9:     Compute the acceptance rate:

$$r = \min\left\{\exp\left\{\frac{1}{\eta} \cdot \left(\sum_{t=1}^T I_{\mathcal{U}_t(\gamma)}(\boldsymbol{u}') - \sum_{t=1}^T I_{\mathcal{U}_t(\gamma)}(\boldsymbol{u}^{(k-1)})\right)\right\}, 1\right\} \tag{3}$$

10:     Set

$$\boldsymbol{u}^{(k)} = \begin{cases} \boldsymbol{u}', & \text{w.p. } r \\ \boldsymbol{u}^{(k-1)}, & \text{w.p. } 1-r \end{cases}$$

11: **end for**
12: Output: $\boldsymbol{u}^{(K)}$

---

Algorithm 2 presents a simulated annealing algorithm to solve the optimization problem $\text{OPT}_\delta(\gamma)$ in Section 4. It takes a similar MCMC routine as Algorithm 1 and we use the same Gaussian random perturbation for the proposal distribution $\mathbb{Q}$. As the temperature parameter $\eta$ decreases, the sampling distribution in Algorithm 2 will gradually be more concentrated on the optimal solution set of $\text{OPT}_\delta(\gamma)$. Algorithm 2 can be implemented more efficiently than Algorithm 1 in that the likelihood ratio calculation in (3) is analytical.

Table 1 reports some numerical results for the two algorithms. For both the Gaussian and $\delta$-corruption settings, we consider three distributions of $\mathcal{P}_{\boldsymbol{a},b}$: (i) a uniform distribution where $\boldsymbol{a} \sim \text{Unif}([1,2]^n)$ and $b \sim \text{Unif}([1,n])$; (ii) a discrete distribution where $\text{Unif}(\{1,2\}^n)$ and $b \sim \text{Unif}(1,...,n)$; (iii) a fixed-$\boldsymbol{a}$ distribution where $\boldsymbol{a} = (1,...,1)^\top$ and $b \sim \text{Unif}(1,...,n)$. For the Gaussian case, the true parameters $(\boldsymbol{\mu}^*, \kappa^*)$ are uniformly generated from $\mathcal{S}^{n-1} \times [1,10]$, and the accuracy is calculated by $(\boldsymbol{\mu}^* \boldsymbol{x}^* - \boldsymbol{\mu}^* \tilde{\boldsymbol{x}}^*)/\boldsymbol{\mu}^* \boldsymbol{x}^*$ where $\boldsymbol{x}^*$ and $\tilde{\boldsymbol{x}}^*$ are defined in Corollary 2. For the Gaussian case, $\boldsymbol{u}^*$ is uniformly generated from $\mathcal{S}^{n-1}$ and $\delta$ is set to be 0.1, and the accuracy is calculated by $\text{Acc}(\hat{\boldsymbol{u}})/\text{Acc}(\boldsymbol{u}^*)$ where $\text{Acc}(u)$ is defined in Section 4. The numbers in Table 1 are reported based on an average of 20 simulation trials, and we run both Algorithm 1 and Algorithm 2 for $K = 1000$ iterations.

We make the following observations from the numerical experiments. First, we remark that the theoretical results in the previous sections provide strong guarantees on the convergence property of the posterior distribution. So the deterioration of the algorithm performance for the case when $n = 25$ is solely caused by the inaccuracy of the approximate sampling in either Algorithm 1 or Algorithm 2. Such inaccuracy can definitely be mitigated to some extent by a more efficient algorithm implementation such as parallel computing. However, we argue that the performance deterioration as $n$ grows may point to a curse of dimensionality that is intrinsic to this estimation problem. Essentially, we aim to estimate a high-dimensional distribution only through partial information, i.e., the sets $\mathcal{U}_t$'s. On the positive end, the algorithms work well for $n \le 10$, so if the learner has the power of choosing $(\boldsymbol{a}_t, b_t)$, s/he can break up the high-dimensional estimation problem into a number of low-dimensional estimation problems by focusing on a handful of dimensions each time. Moreover, we provide a visualization of the condition (1) in Figure 2 for $n = 5$ calculated based on simulation. The visualization supports the existence of $L$ and thus the identifiability of the true parameters when the posterior sampling can be accurately fulfilled.

|  |  | $n=3$ | $n=5$ | $n=10$ | $n=25$ |
|---|---|---|---|---|---|
| | (i) | 99.9% | 99.9% | 98.8% | 59.9% |
| Gaussian | (ii) | 99.9% | 99.3% | 96.9% | 56.9% |
| | (iii) | 99.9% | 94.8% | 92.6% | 68.5% |
| | (i) | 99.9% | 96.7% | 96.0% | 55.7% |
| $\delta$-corru. | (ii) | 99.6% | 97.9% | 97.6% | 63.1% |
| | (iii) | 99.9% | 98.7% | 87.1% | 58.7% |

Table 1: Predictive accuracies under two settings.

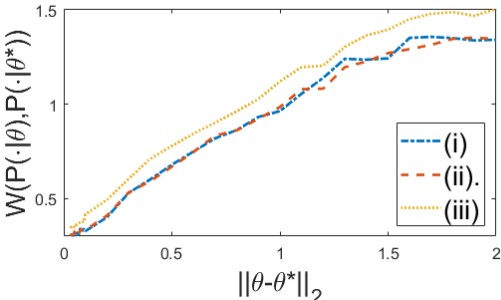

Figure 2: Visualization of (1).

We conclude our discussion with the following remarks.

Query-based model with learner-chosen $(\boldsymbol{a}_t, b_t)$: In this paper, we have focused on the case where the constraints $(\boldsymbol{a}_t, b_t)$'s are stochastically generated. When the concentration parameter $\kappa$ is known for the Gaussian case, there is an efficient way of learning $\boldsymbol{\mu}$ through choosing $(\boldsymbol{a}_t, b_t)$'s (See the Appendix). In addition, the numerical experiments above also inspire a method that dismantles the high-dimensional estimation problem into a number of low-dimensional problems. Another

interesting and important question is whether there exist designs of $(\boldsymbol{a}_t, b_t)$'s such that the posterior sampling can be more efficiently carried out.

Multiple constraints and nonlinear utility: The results in this paper are presented under the setting of a linear objective and a single constraint. We emphasize that the results can be easily generalized to the case of multiple constraints and parameterized nonlinear utility. Thus our result can be viewed as a preliminary effort to address the problem of stochastic inverse optimization. Our conjecture is that when the set $\mathcal{U}_t$ corresponds to a multiple-constraint problem, it may feature more structure and thus facilitate the learning of the utility distribution.

Choice modeling: The stochastic utility model in our paper also draws an interesting connection with the literature on choice modeling, which is a pillar for the pricing and assortment problems in revenue management [TVRVR04, GT+19]. For most of the existing choice models, the learning problem can be viewed as a special case of our study by letting $\boldsymbol{a} = (1, ..., 1)^\top$ and $b = 1$. The results in our paper complement to this line of literature in developing a model where customers can make multiple purchases.

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
