# A  Auxiliary Lemmas

We present some preliminary lemmas in this section. Most of them are basic inequalities in Information Theory, so they are only for auxiliary purposes in our proofs of the main theorems and the corollaries.

**Lemma 2** (Pinsker's inequality). *For any two distributions $\mathcal{P}_1$ and $\mathcal{P}_2$,*

$$D_{TV}(\mathcal{P}_1, \mathcal{P}_2) \le \sqrt{\frac{1}{2} D_{KL}(\mathcal{P}_1, \mathcal{P}_2)},$$

*where $D_{TV}(\cdot, \cdot)$ denotes the total variation distance between two distributions, and $D_{KL}(\cdot, \cdot)$ denotes the KL-divergence between distributions.*

*Proof.* We refer to Lemma 6.2 of the book Gray (2011). $\qquad\square$

**Lemma 3** (Data processing inequality). *Let $\{K_\lambda\}_{\lambda \in \mathcal{X}}$ be a set of random variables indexed by parameter $\lambda$ in some space $\mathcal{X}$. Consider two random variables $\Lambda_1, \Lambda_2$ taking values in $\mathcal{X}$. The following inequality holds*

$$D_{KL}(K_{\Lambda_1}, K_{\Lambda_2}) \le D_{KL}(\Lambda_1, \Lambda_2),$$

*where $D_{KL}(\cdot, \cdot)$ denotes the KL-divergence between two distributions. Here $\{K_\lambda\}_{\lambda \in \mathcal{X}}$ is also called a Markov kernel, a transition probability distribution, or a statistical kernel.*

*Proof.* We refer to Theorem 14 of the article Liese and Vajda (2006). $\qquad\square$

**Lemma 4** (Packing number). *Let $\mathcal{B}_r$ denote the ball in $\mathbb{R}^n$ centered at original point with radius $r$. Then, the $\epsilon$-packing number of $\mathcal{B}_r$ is bounded by*

$$\left(1 + \frac{2r}{\epsilon}\right)^n.$$

*In other words, there exist at most $\left(1 + \frac{2r}{\epsilon}\right)^n$ disjoint $\frac{\epsilon}{2}$ balls in $\mathcal{B}_r$.*

*Proof.* Assume that there are $M$ disjoint $\epsilon/2$-balls. Then, the total volume of those $M$ balls cannot be larger than the volume of a $r + \epsilon/2$ ball, i.e.,

$$M \cdot \left(\frac{\epsilon}{2}\right)^n \le (r + \frac{\epsilon}{2})^n,$$

which implies

$$M \le \left(1 + \frac{2r}{\epsilon}\right)^n.$$

$\qquad\square$

**Lemma 5** (Hoeffding's inequality). *Let $X_1, ..., X_T$ be independent random variables such that $X_t$ takes its values in $[u_t, v_t]$ almost surely for all $t \le T$. Then for every $s > 0$,*

$$\mathbb{P}\left(\left|\frac{1}{T}\sum_{t=1}^{n} X_t - \mathbb{E}X_t\right| \ge s\right) \le 2\exp\left(-\frac{2T^2 s^2}{\sum_{i=1}^{n}(u_t - v_t)^2}\right).$$

*Proof.* We refer to Chapter 2 of the book (Boucheron et al., 2013). $\qquad\square$

**Lemma 6** (Doob's consistency theorem). *Suppose that $\mathbb{P}(\cdot|\boldsymbol{\theta}) \ne \mathbb{P}(\cdot|\boldsymbol{\theta}')$ whenever $\boldsymbol{\theta} \ne \boldsymbol{\theta}'$. Then, for every prior probability measure on the parameter space, the sequence of posterior measures converges to the point mass distribution of the true parameter in distribution for almost every $\boldsymbol{\theta}$.*

*Proof.* We refer to 10.10 of the book Van der Vaart (2000). □

**Lemma 7.** *Let $\mathcal{F}$ be a set of functions whose domain is the support of the distribution of $(\boldsymbol{x}^*, \boldsymbol{a}, b)$. For any probability distribution $\tilde{\mathbb{P}}$ on $\mathcal{F}$, the following inequality holds for all $f \in \mathcal{F}$ and all distributions $\tilde{\mathbb{Q}}$ on $\mathcal{F}$ simultaneously*

$$\mathbb{E}_{\tilde{\mathbb{Q}}}\left[\mathbb{E}[f(\boldsymbol{x}^*, \boldsymbol{a}, b)]\right] \leq \mathbb{E}_{\tilde{\mathbb{Q}}}\left[\sum_{t=1}^{T} f(\boldsymbol{x}_t^*, \boldsymbol{a}_t, b_t)\right] + \sqrt{\frac{D_{KL}(\tilde{\mathbb{P}}, \tilde{\mathbb{Q}}) + \log\frac{T}{\epsilon} + 2}{2T - 1}}.$$

*with probability no less than $1 - \epsilon$. Here the inner expectation on the left-hand-side is taken with respect to $(\boldsymbol{x}^*, \boldsymbol{a}, b)$.*

*Proof.* We refer to Theorem 1 of the article McAllester (2003). □

The following lemma provides a useful bound for the modified Bessel function of the first kind. This function is closely related to the density of the von Mises–Fisher distribution. Typically, the modified Bessel function of the first kind is denoted by $I_\nu(x)$ with the parameter $\nu$. In this paper, to distinguish between the indicator function and this modified Bessel function, we denote the modified Bessel function by $\tilde{I}_\nu(x)$.

**Lemma 8.** *For all $0 < x < y$ and $\nu > 0$,*

$$e^{x-y}\left(\frac{x}{y}\right)^\nu \leq \frac{\tilde{I}_\nu(x)}{\tilde{I}_\nu(y)} \leq e^{y-x}\left(\frac{x}{y}\right)^\nu,$$

*where $\tilde{I}_\nu(\cdot)$ denotes the modified Bessel function of the first kind.*

*Proof.* We refer to Chapter 2 of the article Baricz (2010). □

**Lemma 9.** *Let $\mathcal{F}$ be a class of functions $f : X \to [a, b]$, and $\{X_t\}_{t=1}^{T}$ be i.i.d. random variables taking values on $X$. Then for every $s > 0$*

$$\mathbb{P}\left(\sup_{f \in \mathcal{F}}\left|\frac{1}{T}\sum_{t=1}^{T} f(X_t) - \mathbb{E}[f(X_1)]\right| \leq \mathbb{E}\left[\sup_{f \in \mathcal{F}}\left|\frac{1}{T}\sum_{t=1}^{T}\sigma_t f(X_t)\right|\right] + s\right) \leq \exp\left(\frac{2Ts^2}{(b-a)^2}\right),$$

*where $\{\sigma_t\}_{t=1}^{T}$ denotes a set of i.i.d. random signs satisfying $\mathbb{P}(\sigma_t = 1) = \mathbb{P}(\sigma_t = -1) = \frac{1}{2}$.*

*Proof.* □

# B   Proof of Section 3

In this section, we provide the proofs of Section 3 and analyze the convergence of the posterior distribution. As we mentioned earlier, the Wasserstein distance in Theorem 1 is not essential, and all other equivalent metrics or weaker metrics are also valid, such as the total variation distance, the Hellinger distance and the Prokhorov metric (See Gibbs and Su (2002) for more about the relationships between different probability metrics). In this section, we show that Theorem 1 holds with respect to the total variation distance, and then explain how to obtain the result for the Wasserstein distance from there. The main idea is inspired by Ghosal et al. (2000). In this section, we first state and show several key lemmas and then prove the main theorem.

## B.1 Key Lemmas for the Proof of Theorem 1

Recall $\mathcal{D}_T = \{(\boldsymbol{x}_t^*, \boldsymbol{a}_t, b_t)\}_{t=1}^T$ is the data set, and $\mathbb{P}((\boldsymbol{x}^*, \boldsymbol{a}, b)|\boldsymbol{\theta})$ is the likelihood distribution under the parameter $\boldsymbol{\theta}$. In the following, we denote the parameter space $\mathcal{S}^{n-1} \times (\underline{\kappa}, \bar{\kappa})$ by $\Theta$, denote the total variation distance between $\mathbb{P}((\boldsymbol{x}^*, \boldsymbol{a}, b)|\boldsymbol{\theta}_1)$ and $\mathbb{P}((\boldsymbol{x}^*, \boldsymbol{a}, b)|\boldsymbol{\theta}_2)$ by $D_{TV}(\boldsymbol{\theta}_1, \boldsymbol{\theta}_2)$, where $\boldsymbol{\theta}_1, \boldsymbol{\theta}_2 \in \Theta$ are two parameters, and denote the $\epsilon$-packing number of a parameter subset $\tilde{\Theta} \subset \mathcal{S} \times (\underline{\kappa}, \bar{\kappa})$ with a metric $D_{metric}$ by $\mathcal{N}\left(\epsilon, \tilde{\Theta}, D_{metric}\right)$. For example, if the underlying metric is the total variation metric, the corresponding $\epsilon$-packing number is denoted by $\mathcal{N}\left(\epsilon, \tilde{\Theta}, D_{TV}\right)$.

**Lemma 10** (Lemma 7.1 in (Ghosal et al., 2000)). *Suppose the $\epsilon/2$-packing number of the parameter space $\Theta$ with the total variation distance is bounded by some positive constant $C$, i.e.,*

$$\mathcal{N}\left(\frac{\epsilon}{2}, \Theta, D_{TV}\right) \leq C.$$

*Then, for $t, j \in \mathbb{N}$, there exist a function $\phi_t$, which maps a dataset with $t$ samples $\mathcal{D}_t$ to $[0, 1]$, such that*

$$\mathbb{E}_{\boldsymbol{\theta}^*}[\phi_t] \leq \frac{C \exp(-2t\epsilon^2)}{1 - \exp(-2t\epsilon^2)}, \tag{4}$$

$$\sup_{D_{TV}(\boldsymbol{\theta}, \boldsymbol{\theta}^*) > j\epsilon} \mathbb{E}_{\boldsymbol{\theta}}[1 - \phi_t] \leq \exp(-2tj^2\epsilon^2), \tag{5}$$

*where the expectation is taken with respect to the underlying dataset $\mathcal{D}_t$ under a distribution specified by the corresponding parameter.*

*Proof.* We refer to Lemma 7.1 in Ghosal et al. (2000). $\square$

Intuitively, the lemma states that if the packing number is bounded, we can find a function $\phi_t$ such that its expectation is close to 0 when taken under a distribution with the true parameter, and it is close to 1 otherwise. In statistics, the function $\phi_t$ naturally serves as a test. In the proof of Theorem 1, we will see that this test function $\phi_t$ plays an important role in bounding the numerator of the posterior distribution.

To apply this lemma for the proof of Theorem 1, we first bound the packing number to meet the lemma's condition. Here $D_{KL}(\cdot, \cdot)$ denotes the KL divergence between two distributions.

**Lemma 11.** *For any two parameters $\boldsymbol{\theta}_1 = (\boldsymbol{\mu}_1, \kappa_1)$ and $\boldsymbol{\theta}_2 = (\boldsymbol{\mu}_2, \kappa_2)$ in $\Theta := \mathcal{S}^{n-1} \times (\underline{\kappa}, \bar{\kappa})$,*

$$D_{TV}(\boldsymbol{\theta}_1, \boldsymbol{\theta}_2) \leq \sqrt{\frac{1}{2} D_{KL}(\mathbb{P}((\boldsymbol{x}^*, \boldsymbol{a}, b)|\boldsymbol{\theta}_1), \mathbb{P}((\boldsymbol{x}^*, \boldsymbol{a}, b)|\boldsymbol{\theta}_2))} \leq \sqrt{\|\boldsymbol{\theta}_1 - \boldsymbol{\theta}_2\|_2}. \tag{6}$$

*Furthermore, we have*

$$\mathcal{N}(\epsilon, \Theta, D_{TV}) \leq \left(1 + \frac{\max(2, 2\bar{\kappa})}{\epsilon^2}\right)^n.$$

*Proof.* The first inequality comes directly from Pinsker's inequality (Lemma 2). From Lemma 3, we have

$$D_{KL}(\mathbb{P}((\boldsymbol{x}^*, \boldsymbol{a}, b)|\boldsymbol{\theta}_1), \mathbb{P}((\boldsymbol{x}^*, \boldsymbol{a}, b)|\boldsymbol{\theta}_2))$$

$$\leq D_{KL}(\mathcal{P}_{\boldsymbol{u}}(\cdot|\boldsymbol{\theta}_1), \mathcal{P}_{\boldsymbol{u}}(\cdot|\boldsymbol{\theta}_2))$$

$$= \int_{\boldsymbol{u} \in \mathcal{S}^{n-1}} C_n(\kappa_1) \exp(\kappa_1 \boldsymbol{\mu}_1^\top \boldsymbol{u}) \cdot \log\left(\frac{C_n(\kappa_1) \exp(\kappa_1 \boldsymbol{\mu}_1^\top \boldsymbol{u})}{C_n(\kappa_2) \exp(\kappa_2 \boldsymbol{\mu}_2^\top \boldsymbol{u})}\right)$$

$$= \int_{\boldsymbol{u} \in \mathcal{S}^{n-1}} C_n(\kappa_1) \exp(\kappa_1 \boldsymbol{\mu}_1^\top \boldsymbol{u}) \cdot \left((\kappa_1 \boldsymbol{\mu}_1 - \kappa_2 \boldsymbol{\mu}_2)^\top \boldsymbol{u} + \log\left(\frac{\kappa_1^{n/2-1} \tilde{I}_{n/2-1}(\kappa_2)}{\kappa_2^{n/2-1} \tilde{I}_{n/2-1}(\kappa_1)}\right)\right)$$

$$\leq \int_{\boldsymbol{u} \in \mathcal{S}^{n-1}} C_n(\kappa_1) \exp(\kappa_1 \boldsymbol{\mu}_1^\top \boldsymbol{u}) \cdot \left((\kappa_1 \boldsymbol{\mu}_1 - \kappa_2 \boldsymbol{\mu}_2)^\top \boldsymbol{u} + |\kappa_1 - \kappa_2|\right)$$

$$\leq 2\|\kappa_1 \boldsymbol{\mu}_1 - \kappa_2 \boldsymbol{\mu}_2\|_2,$$

where

$$C_n(\kappa) = \left(\int_{\boldsymbol{u} \in \mathcal{S}^{n-1}} \exp(\kappa \boldsymbol{\mu}^\top \boldsymbol{u})\right)^{-1} = \frac{\kappa^{n/2-1}}{(2\pi)^{n/2} \tilde{I}_{n/2-1}(\kappa)},$$

which is independent of the choice of $\boldsymbol{\mu} \in \mathcal{S}^{n-1}$, and $\tilde{I}_{n/2-1}(\cdot)$ denotes the modified Bessel functions of the first kind. Here, the first line comes from Lemma 3. The second line comes from the definition of the KL-divergence and the density function of the von Mises–Fisher distribution, and the third line comes from the definition of $C_n(\kappa)$. The fourth line comes from Lemma 8 that

$$\left|\log \frac{\kappa_1^{n/2-1} \tilde{I}_{n/2-1}(\kappa_2)}{\kappa_2^{n/2-1} \tilde{I}_{n/2-1}(\kappa_1)}\right| \leq \left|\max(\log e^{\kappa_1 - \kappa_2}, \log e^{\kappa_2 - \kappa_1})\right| = |\kappa_1 - \kappa_2|,$$

and the last line comes from Cauchy inequality that

$$(\kappa_1 \boldsymbol{\mu}_1 - \kappa_2 \boldsymbol{\mu}_2)^\top \boldsymbol{u} \leq \|\kappa_1 \boldsymbol{\mu}_1 - \kappa_2 \boldsymbol{\mu}_2\|_2 \|\boldsymbol{u}\|_2 = \|\kappa_1 \boldsymbol{\mu}_1 - \kappa_2 \boldsymbol{\mu}_2\|_2.$$

Therefore, the $\epsilon^2$-ball centered at some $\tilde{\boldsymbol{\theta}}$ with the L$_2$-norm is a subset of the $\epsilon$-ball centered at $\tilde{\boldsymbol{\theta}}$ with the total variation metric, i.e.,

$$\{\boldsymbol{\theta} : \|\boldsymbol{\theta} - \tilde{\boldsymbol{\theta}}\|_2 \leq \epsilon^2\} \subset \{\boldsymbol{\theta} : D_{TV}(\boldsymbol{\theta}, \tilde{\boldsymbol{\theta}}) \leq \epsilon\}.$$

Thus, the $\epsilon$-packing number of $\Theta$ with the total variance metric is bounded by the $\epsilon^2$-packing number of $\Theta$ with the L$_2$-norm, that is,

$$\mathcal{N}(\epsilon, \Theta, D_{TV}) \leq \mathcal{N}(\epsilon^2, \Theta, \|\cdot\|_2).$$

Finally, by Lemma 4, for all $\epsilon < 1$, we have

$$\mathcal{N}(\epsilon, \Theta, D_{TV}) \leq \left(1 + \frac{\max(2, 2\bar{\kappa})}{\epsilon^2}\right)^n.$$

$\square$

We will use the above two lemmas to bound the numerator of the posterior distribution. First, we derive a lower bound of the denominator of the posterior distribution. And then we will combine those two parts to establish the statement of the posterior distribution.

**Lemma 12.** *For any $\epsilon > 0$, the following inequality holds with probability no less than $1 - \exp(-T\epsilon^2/4\bar{\kappa}^2)$,*

$$\int_\Theta \frac{\mathbb{P}(\mathcal{D}_T|\boldsymbol{\theta})}{\mathbb{P}(\mathcal{D}_T|\boldsymbol{\theta}^*)}\mathbb{P}_0(\mathrm{d}\boldsymbol{\theta}) \geq \left(\frac{2\epsilon^2}{\bar{\kappa}}\right)^n \cdot \exp(-T\epsilon^2) \tag{7}$$

*Proof.* Let

$$\Theta_{KL}(\epsilon) := \left\{\boldsymbol{\theta} \in \Theta : D_{KL}(\mathbb{P}((\boldsymbol{x}^*, \boldsymbol{a}, b)|\boldsymbol{\theta}^*), \mathbb{P}((\boldsymbol{x}^*, \boldsymbol{a}, b)|\boldsymbol{\theta})) \leq \epsilon^2\right\}.$$

By inequality (6) in Lemma 11, we have that $\Theta_{KL}(\epsilon)$ contains the $2\epsilon^2$-ball centered at $\boldsymbol{\theta}^*$ with the 2-norm and, therefore,

$$\mathbb{P}_0(\Theta_{KL}(\epsilon)) \geq \left(\frac{2\epsilon^2}{\bar{\kappa}}\right)^n. \tag{8}$$

Next, we show that with probability no less than $1 - \exp(-T\epsilon^2/4\bar{\kappa}^2)$,

$$\int_{\Theta_{KL}(\epsilon)} \frac{\mathbb{P}(\mathcal{D}_T|\boldsymbol{\theta})}{\mathbb{P}(\mathcal{D}_T|\boldsymbol{\theta}^*)}\mathbb{P}_0(\mathrm{d}\boldsymbol{\theta}) \geq \exp\left(-T\epsilon^2\right) \cdot \mathbb{P}_0(\Theta_{KL}(\epsilon)). \tag{9}$$

Since the range of the density function of $\mathcal{P}_u(\cdot|\boldsymbol{\theta})$ is between $[\mathrm{e}^{-\bar{\kappa}}, \mathrm{e}^{\bar{\kappa}}]$ for all $\boldsymbol{\theta} \in \Theta$, we have that

$$-2\bar{\kappa} \leq \log \frac{\mathbb{P}((\boldsymbol{x}^*, \boldsymbol{a}, b)|\boldsymbol{\theta})}{\mathbb{P}((\boldsymbol{x}^*, \boldsymbol{a}, b)|\boldsymbol{\theta}^*)} \leq 2\bar{\kappa}$$

for all $(\boldsymbol{x}^*, \boldsymbol{a}, b)$. Then, by Hoeffding's inequality (Lemma 5), we have, with probability no more than $\exp(-T\epsilon^2/4\bar{\kappa}^2)$,

$$\sum_{t=1}^T \int_{\Theta_{KL}(\epsilon)} \log \frac{\mathbb{P}((\boldsymbol{x}_t^*, \boldsymbol{a}_t, b_t)|\boldsymbol{\theta})}{\mathbb{P}((\boldsymbol{x}_t^*, \boldsymbol{a}_t, b_t)|\boldsymbol{\theta}^*)}\mathbb{P}_0(\mathrm{d}\boldsymbol{\theta}) \leq T\mathbb{E}_{\boldsymbol{\theta}^*}\left(\int_{\Theta_{KL}(\epsilon)} \log \frac{\mathbb{P}((\boldsymbol{x}_t^*, \boldsymbol{a}_t, b_t)|\boldsymbol{\theta})}{\mathbb{P}((\boldsymbol{x}_t^*, \boldsymbol{a}_t, b_t)|\boldsymbol{\theta}^*)}\mathbb{P}_0(\mathrm{d}\boldsymbol{\theta})\right) - 2T\epsilon^2$$

$$= T\int_{\Theta_{KL}(\epsilon)} D_{KL}(\mathbb{P}((\boldsymbol{x}^*, \boldsymbol{a}, b)|\boldsymbol{\theta}^*), \mathbb{P}((\boldsymbol{x}^*, \boldsymbol{a}, b)|\boldsymbol{\theta}))\mathbb{P}_0(\mathrm{d}\boldsymbol{\theta}) - 2T\epsilon^2$$

$$\leq T\epsilon^2 - 2T\epsilon^2 = -T\epsilon^2 \tag{10}$$

where the first line comes directly from Hoeffding's inequality, the second line comes from Fubini's theorem, and the last line comes from the definition of $\Theta_{KL}(\epsilon)$. Then, by Jensen's inequality,

$$\log \int_{\Theta_{KL}(\epsilon)} \frac{\mathbb{P}(\mathcal{D}_T|\boldsymbol{\theta})}{\mathbb{P}(\mathcal{D}_T|\boldsymbol{\theta}^*)} \frac{\mathbb{P}_0(\mathrm{d}\boldsymbol{\theta})}{\mathbb{P}_0(\Theta_{KL}(\epsilon))} = \log \int_{\Theta_{KL}(\epsilon)} \prod_{t=1}^T \frac{\mathbb{P}((\boldsymbol{x}_t^*, \boldsymbol{a}_t, b_t)|\boldsymbol{\theta})}{\mathbb{P}((\boldsymbol{x}_t^*, \boldsymbol{a}_t, b_t)|\boldsymbol{\theta}^*)} \frac{\mathbb{P}_0(\mathrm{d}\boldsymbol{\theta})}{\mathbb{P}_0(\Theta_{KL}(\epsilon))} \tag{11}$$

$$\geq \sum_{t=1}^T \int_{\Theta_{KL}(\epsilon)} \log \frac{\mathbb{P}((\boldsymbol{x}_t^*, \boldsymbol{a}_t, b_t)|\boldsymbol{\theta})}{\mathbb{P}((\boldsymbol{x}_t^*, \boldsymbol{a}_t, b_t)|\boldsymbol{\theta}^*)} \frac{\mathbb{P}_0(\mathrm{d}\boldsymbol{\theta})}{\mathbb{P}_0(\Theta_{KL}(\epsilon))},$$

Finally, combining (10) with (11), we have with probability no less than $1 - \exp(-T\epsilon^2/4\bar{\kappa}^2)$,

$$\int_\Theta \frac{\mathbb{P}(\mathcal{D}_T|\boldsymbol{\theta})}{\mathbb{P}(\mathcal{D}_T|\boldsymbol{\theta}^*)}\mathbb{P}_0(\mathrm{d}\boldsymbol{\theta}) \geq \int_{\Theta_{KL}(\epsilon)} \frac{\mathbb{P}(\mathcal{D}_T|\boldsymbol{\theta})}{\mathbb{P}(\mathcal{D}_T|\boldsymbol{\theta}^*)}\mathbb{P}_0(\mathrm{d}\boldsymbol{\theta})$$

$$\geq \exp\left(-T\epsilon^2\right) \cdot \mathbb{P}_0(\Theta_{KL}(\epsilon))$$

$$\geq \left(\frac{2\epsilon^2}{\bar{\kappa}}\right)^n \cdot \exp(-T\epsilon^2),$$

where the first inequality is obtained by the non-negativity of the integrand, the second inequality comes from (9), and the last line comes from (8). $\square$

## B.2 Proof of Theorem 1

In this part, we combine three lemmas in the previous section and show Theorem 1.

*Proof.* Let

$$\epsilon_T = \max(4, 4\bar{\kappa}) \frac{\sqrt{n} \cdot \log T}{T^{1/2}}.$$

By Lemma 12, we have, with probability no less than $1 - \exp(-T\epsilon^2/4\bar{\kappa}^2)$, the inequality

$$\int_\Theta \frac{\mathbb{P}(\mathcal{D}_T|\boldsymbol{\theta})}{\mathbb{P}(\mathcal{D}_T|\boldsymbol{\theta}^*)} \mathbb{P}_0(\mathrm{d}\boldsymbol{\theta}) \geq \left(\frac{2\epsilon_T^2}{\bar{\kappa}}\right)^n \cdot \exp(-T\epsilon_T^2) \geq \exp(-2T\epsilon_T^2) \tag{12}$$

holds for all $T$ satisfying $T \geq n \log T$. In (16), we will use this inequality to establish a lower bound for the denominator of the posterior distribution.

By Lemma 11, we have for all $T \geq 3$

$$\mathcal{N}\left(\frac{\epsilon_T}{2}, \Theta, D_{TV}\right) \leq \left(\frac{5T}{16n \cdot \log T}\right)^n \leq \exp(T\epsilon_T^2),$$

which gives an upper bound of the packing number and thus verifies the condition of Lemma 10. Then, by Lemma 10, for all $T \geq 4$, there exists a function $\phi_T$ mapping the data set $\mathcal{D}_T$ to $[0,1]$, which satisfies

$$\mathbb{E}_{\boldsymbol{\theta}^*}[\phi_T] \leq \frac{\exp(T\epsilon_T^2)\exp(-2T\epsilon_T^2)}{1 - \exp(-2T\epsilon_T^2)} \leq 2\exp(-T\epsilon_T^2), \tag{13}$$

$$\sup_{D_{TV}(\boldsymbol{\theta},\boldsymbol{\theta}^*)>2\epsilon_T} \mathbb{E}_{\boldsymbol{\theta}}[1 - \phi_T] \leq \exp(-4T\epsilon_T^2). \tag{14}$$

Let

$$\tilde{\Theta}_T := \left\{\boldsymbol{\theta} \in \Theta : D_{TV}\left(\mathbb{P}\left((\boldsymbol{x}^*, \boldsymbol{a}, b)|\boldsymbol{\theta}\right), \mathbb{P}\left((\boldsymbol{x}^*, \boldsymbol{a}, b)|\boldsymbol{\theta}^*\right)\right) \leq \max\left(8, 8\bar{\kappa}\right) \frac{\sqrt{n} \cdot \log T}{T^{1/2}}\right\}$$

We then have

$$\mathbb{E}_{\boldsymbol{\theta}^*}\left[(1 - \phi_T)\int_{\tilde{\Theta}_T^c} \prod_{t=1}^T \frac{\mathbb{P}(\mathcal{D}_T|\boldsymbol{\theta})}{\mathbb{P}(\mathcal{D}_T|\boldsymbol{\theta}^*)}\mathbb{P}_0(\mathrm{d}\boldsymbol{\theta})\right] = \int_{\tilde{\Theta}_T^c} \mathbb{E}_{\boldsymbol{\theta}^*}\left[(1 - \phi_T)\prod_{t=1}^T \frac{\mathbb{P}(\mathcal{D}_T|\boldsymbol{\theta})}{\mathbb{P}(\mathcal{D}_T|\boldsymbol{\theta}^*)}\right]\mathbb{P}_0(\mathrm{d}\boldsymbol{\theta})$$

$$= \int_{\tilde{\Theta}_T^c} \mathbb{E}_{\boldsymbol{\theta}}(1 - \phi_T)\mathbb{P}_0(\mathrm{d}\boldsymbol{\theta}) \tag{15}$$

$$\leq \exp(-4T\epsilon_T^2),$$

where the first line is obtained by Fubini's theorem, the second line is obtained directly by computing the inner integral, and the last line comes from the definition of $\tilde{\Theta}_T$ and inequality (14). Denote the low probability event corresponding to inequality (12) as $\mathcal{E}_T$. By combining (12) and (15), we have

$$\mathbb{E}_{\boldsymbol{\theta}^*}\left[\mathbb{P}_T(\tilde{\Theta}_T^c)(1 - \phi_T)I_{\mathcal{E}_T}\right] = \mathbb{E}_{\boldsymbol{\theta}^*}\left[\frac{(1 - \phi_T)I_{A_T}\int_{\tilde{\Theta}_T^c} \frac{\mathbb{P}(\mathcal{D}_T|\boldsymbol{\theta})}{\mathbb{P}(\mathcal{D}_T|\boldsymbol{\theta}^*)}\mathbb{P}(\mathrm{d}\boldsymbol{\theta})}{\int_\Theta \frac{\mathbb{P}(\mathcal{D}_T|\boldsymbol{\theta})}{\mathbb{P}(\mathcal{D}_T|\boldsymbol{\theta}^*)}\mathbb{P}(\mathrm{d}\boldsymbol{\theta})}\right]$$

$$\leq \exp(-4T\epsilon_T^2)\exp(2T\epsilon_T^2) = \exp(-2T\epsilon_T^2), \tag{16}$$

where the first equality comes from the definition of the posterior distribution, and the second line is obtained by plugging in (12) and (15).

Finally, we have

$$\mathbb{E}_{\boldsymbol{\theta}^*}\left[\mathbb{P}_T(\tilde{\Theta}_T)\right] \geq 1 - \mathbb{E}_{\boldsymbol{\theta}^*}\left[(1-\phi_T)I_{\mathcal{E}_T}\mathbb{P}_T(\tilde{\Theta}_T^c)\right] - \mathbb{E}_{\boldsymbol{\theta}^*}\left[\phi_T\mathbb{P}_T(\tilde{\Theta}_T^c)\right] - \mathbb{E}_{\boldsymbol{\theta}^*}\left[I_{\mathcal{E}_T}\mathbb{P}_T(\tilde{\Theta}_T^c)\right]$$

$$\geq 1 - \mathbb{E}_{\boldsymbol{\theta}^*}\left[(1-\phi_T)I_{\mathcal{E}_T}\mathbb{P}_T(\tilde{\Theta}_T^c)\right] - \mathbb{E}_{\boldsymbol{\theta}^*}\left[\phi_T\right] - \mathbb{E}_{\boldsymbol{\theta}^*}\left[I_{\mathcal{E}_T}\right] \qquad (17)$$

$$\geq 1 - 2\exp(-T\epsilon_T^2) - \exp(-T\epsilon^2/4\bar{\kappa}^2)$$

$$\geq 1 - \frac{3}{T},$$

where the first line comes from the fact that the posterior probability is bounded by 1, the second line comes from (13) and (16), and the last line comes from the definition of $\epsilon_T$. The inequality above also indicates that $\mathbb{P}_T(\tilde{\Theta}_T)$ converges to 1 in $L_1$ norm, which implies the convergence in probability.

To obtained a similar result for the $L_2$ Wasserstein distance, we only need to establish the relationship between $\Theta_T$ and $\tilde{\Theta}_T$. Notice that $\mathcal{P}_{\boldsymbol{a},b}$ is independent of $\mathcal{P}_u(\cdot|\boldsymbol{\theta})$ and the maximum distance between two points in $[0,1]^n$ is no larger than $\sqrt{n}$. We have

$$W_2(\mathcal{P}((\boldsymbol{x}^*,\boldsymbol{a},b)|\boldsymbol{\theta}), \mathcal{P}((\boldsymbol{x}^*,\boldsymbol{a},b)|\boldsymbol{\theta}^*)) \leq \sqrt{n}D_{TV}(\boldsymbol{\theta},\boldsymbol{\theta}^*).$$

Thus, the result of (17) also holds for the $L_2$ Wasserstein distance metric if we have an additional $\sqrt{n}$ factor in the definition of $\epsilon_T$ and $\tilde{\Theta}_T$, which is exactly $\Theta_T$ in Theorem 1. $\qquad \square$

## B.3  Proof of Corollary 1

*Proof.* The convergence of the posterior distribution to the point mass distribution can be directly obtained by Doob's consistency theorem (Lemma 6). Here, we only show the second part, i.e., the upper bound of the posterior expectation $\mathbb{E}_T[\|\boldsymbol{\theta}_T - \boldsymbol{\theta}^*\|_2]$.

From the proof of Theorem 1, we have that for sufficient large $T$

$$\mathbb{E}_{\boldsymbol{\theta}^*}\left[\mathbb{P}_T(\Theta_T^c)\right] \leq \frac{3}{T}.$$

Then, by Markov's inequality, we have

$$\mathbb{P}_{\boldsymbol{\theta}^*}\left(\mathbb{P}_T(\Theta_T^c) > \frac{n\log T}{2LT^{1/2}}\right) \leq \frac{6L}{n \cdot T^{1/2}\log T}. \qquad (18)$$

Next, from the condition (1),

$$\mathcal{W}\left(\mathbb{P}\left((\boldsymbol{x}_t^*,\boldsymbol{a}_t,b)|\boldsymbol{\theta}\right), \mathbb{P}\left((\boldsymbol{x}_t^*,\boldsymbol{a}_t,b)|\boldsymbol{\theta}^*\right)\right) \geq L \cdot \|\boldsymbol{\theta} - \boldsymbol{\theta}^*\|_2.$$

Consequently, for any $\boldsymbol{\theta} \in \Theta_T$, we have

$$\|\boldsymbol{\theta} - \boldsymbol{\theta}^*\|_2 \leq \max(8,8\bar{\kappa}) \frac{n \cdot \log T}{L \cdot T^{1/2}}. \qquad (19)$$

Combining (18) and (19), we have with probability no less than $1 - \frac{6L}{n \cdot T^{1/2}\log T}$

$$\mathbb{E}_T\left[\|\boldsymbol{\theta}_T - \boldsymbol{\theta}^*\|_2\right] \leq \max(8,8\bar{\kappa}) \frac{n \cdot \log T}{L \cdot T^{1/2}} \cdot \mathbb{P}_T(\Theta_T) + 2\bar{\kappa} \cdot \mathbb{P}_T(\Theta_T^c)$$

$$\leq \max(8,8\bar{\kappa}) \frac{n \cdot \log T}{L \cdot T^{1/2}} + \max(1,1\bar{\kappa}) \frac{n \cdot \log T}{L \cdot T^{1/2}} = \max(9,9\bar{\kappa}) \frac{n \cdot \log T}{L \cdot T^{1/2}}.$$

Here, the first inequality is obtained by the fact the the maximum distance between two different pa-

rameters are bounded by $2\bar{\kappa}$, and the second inequality is obtained by (19). $\qquad\square$

## B.4 Proof of Corollary 2

*Proof.* Recall the definition of $\tilde{\Theta}_T$

$$\tilde{\Theta}_T = \left\{ \boldsymbol{\theta} \in \Theta : D_{TV}\left(\mathbb{P}\left((\boldsymbol{x}^*, \boldsymbol{a}, b)|\boldsymbol{\theta}\right), \mathbb{P}\left((\boldsymbol{x}^*, \boldsymbol{a}, b)|\boldsymbol{\theta}^*\right)\right) \le \max\left(8, 8\bar{\kappa}\right) \frac{\sqrt{n} \cdot \log T}{T^{1/2}} \right\}.$$

In a similar way as (18), we obtain

$$\mathbb{P}_{\boldsymbol{\theta}^*}\left(\mathbb{P}_T(\tilde{\Theta}_T^c) > \frac{\sqrt{n}\log T}{2T^{1/2}}\right) \le \frac{6}{\sqrt{n} \cdot T^{1/2}\log T}.$$

Then we utilize the total variation distance to bound the difference between optimal solutions. From the integral representation of the total variation distance, we have, for any $\boldsymbol{\theta}$,

$$D_{TV}\left(\mathbb{P}\left((\boldsymbol{x}^*, \boldsymbol{a}, b)|\boldsymbol{\theta}\right), \mathbb{P}\left((\boldsymbol{x}^*, \boldsymbol{a}, b)|\boldsymbol{\theta}^*\right)\right) = \frac{1}{2}\int_{(\boldsymbol{x}^*, \boldsymbol{a}, b)}\left|1 - \frac{\mathbb{P}\left((\boldsymbol{x}^*, \boldsymbol{a}, b)|\boldsymbol{\theta}^*\right)}{\mathbb{P}\left((\boldsymbol{x}^*, \boldsymbol{a}, b)|\boldsymbol{\theta}\right)}\right|\mathbb{P}\left((\mathrm{d}\boldsymbol{x}^*, \mathrm{d}\boldsymbol{a}, \mathrm{d}b)|\boldsymbol{\theta}\right). \quad (20)$$

On the right hand side, the ratio $\frac{\mathbb{P}((\boldsymbol{x}^*, \boldsymbol{a}, b)|\boldsymbol{\theta})}{\mathbb{P}((\boldsymbol{x}^*, \boldsymbol{a}, b)|\boldsymbol{\theta}^*)}$ is the probability that the optimal solutions (corresponding to $\boldsymbol{\theta}^*$ and $\boldsymbol{\theta}$) coincide for a fixed pair of $(\boldsymbol{a}, b)$. Thus, the integration calculates the probability that the the optimal solution corresponding to $\boldsymbol{\theta}$ is different from the optimal solution corresponding to $\boldsymbol{\theta}^*$. Let $\boldsymbol{\theta}_T$ be a random parameter drawn from the posterior distribution. Denote $\tilde{\boldsymbol{x}}^*$ as the optimal solution corresponding to $\boldsymbol{\theta}_T$ given $(\boldsymbol{a}, b)$, and $\boldsymbol{x}^*$ as the optimal solution corresponding to $\boldsymbol{\theta}^*$ given $(\boldsymbol{a}, b)$. We have

$$\mathbb{P}_T(\boldsymbol{x}^* \ne \tilde{\boldsymbol{x}}^*) \le 2D_{TV}\left(\mathbb{P}\left((\boldsymbol{x}^*, \boldsymbol{a}, b)|\boldsymbol{\theta}_T\right), \mathbb{P}\left((\boldsymbol{x}^*, \boldsymbol{a}, b)|\boldsymbol{\theta}^*\right)\right). \quad (21)$$

Thus, with probability no less than $1 - \frac{6}{\sqrt{n} \cdot T^{1/2}\log T}$, we have

$$\begin{aligned}
\mathbb{E}[\|\boldsymbol{x}^* - \tilde{\boldsymbol{x}}^*\|_2] &\le \sqrt{n}\mathbb{P}_T(\boldsymbol{x}^* \ne \tilde{\boldsymbol{x}}^*)\\
&\le 2\sqrt{n}D_{TV}\left(\mathbb{P}\left((\boldsymbol{x}^*, \boldsymbol{a}, b)|\boldsymbol{\theta}\right), \mathbb{P}\left((\boldsymbol{x}^*, \boldsymbol{a}, b)|\boldsymbol{\theta}^*\right)\right),\\
&\le \max\left(16, 16\bar{\kappa}\right)\frac{n \cdot \log T}{T^{1/2}}
\end{aligned}$$

where the first line comes from the fact that the maximum distance between any two solutions is bounded by $\sqrt{n}$, the second line comes from (21), and the last line comes from the definition of $\tilde{\Theta}_T$.

We remark that the statement is also true for general distributions by replacing the probability ratio in (20) with the Radon–Nikodym derivative.

$\qquad\square$

# C Proof of Section 4

In this section, we prove the results in Section 4.

## C.1 Proof of Lemma 1

*Proof.* For an observation of $(\boldsymbol{x}^*, \boldsymbol{a}, b)$, let

$$\mathcal{U} := \left\{\boldsymbol{u} \in \mathcal{S}^{n-1} : \boldsymbol{x}^* \text{ is an optimal solution of } \mathrm{LP}(\boldsymbol{u}, \boldsymbol{a}, b)\right\}.$$

From the LP's optimality conditions, we know that $\boldsymbol{u} \in \mathcal{U}$ if and only if the following three inequalities hold:

$$- \min_{\{i:x_i^* > 0\}} \left( \frac{u_i}{a_i} \right) \leq 0, \tag{22}$$

$$\max_{\{i:x_i^*=0, u_i \neq 0\}} \frac{u_i}{a_i} \leq 0, \text{ if } \boldsymbol{a}^\top \boldsymbol{x}^* < b \tag{23}$$

$$\max_{\{i:x_i^*=0, u_i>0\}} \frac{u_i}{a_i} - \min_{\{i:x_i>0\}} \frac{u_i}{a_i} \leq 0, \text{ if } \boldsymbol{a}^\top \boldsymbol{x}^* = b. \tag{24}$$

All the three inequalities can be expressed by linear constraints, so we finish the proof. $\qquad\square$

## C.2  Proof of Proposition 1

*Proof.* The proof is a direct application of Hoeffding's inequality. Denote $X_t$ as the indicator function of $\boldsymbol{u}_t \neq \boldsymbol{u}^*$. By Hoeffding's inequality, we have

$$\mathbb{P}\left( \frac{1}{T} \sum_{t=1}^{T} X_t \leq \delta + \frac{\log T}{\sqrt{T}} \right) \leq \frac{1}{T}.$$

Then, it is sufficient to show that

$$\max_{\boldsymbol{u} \in \mathcal{S}^{n-1}} \left| \frac{1}{T} \sum_{t=1}^{T} I_{\mathcal{U}_t}(\boldsymbol{u}) - \frac{1}{T} \sum_{t=1}^{T} I_{\bar{\mathcal{U}}_t}(\boldsymbol{u}) \right| \leq \frac{1}{T} \sum_{t=1}^{T} X_t.$$

To see this, we have

$$\max_{\boldsymbol{u} \in \mathcal{S}^{n-1}} \left| \frac{1}{T} \sum_{t=1}^{T} I_{\mathcal{U}_t}(\boldsymbol{u}) - \frac{1}{T} \sum_{t=1}^{T} I_{\bar{\mathcal{U}}_t}(\boldsymbol{u}) \right| \leq \max_{\boldsymbol{u} \in \mathcal{S}^{n-1}} \left| \frac{1}{T} \sum_{\{t:X_t=1\}} I_{\mathcal{U}_t}(\boldsymbol{u}) - \frac{1}{T} \sum_{\{t:X_t=1\}} I_{\bar{\mathcal{U}}_t}(\boldsymbol{u}) \right|$$

$$\leq \max_{\boldsymbol{u} \in \mathcal{S}^{n-1}} \frac{1}{T} \sum_{\{t:X_t=1\}} \left| I_{\mathcal{U}_t}(\boldsymbol{u}) - I_{\bar{\mathcal{U}}_t}(\boldsymbol{u}) \right|$$

$$\leq \max_{\boldsymbol{u} \in \mathcal{S}^{n-1}} \frac{1}{T} \sum_{\{t:X_t=1\}} 1$$

$$= \frac{1}{T} \sum_{t=1}^{T} X_t,$$

where the first inequality is obtained by the fact that $I_{\mathcal{U}_t}(\boldsymbol{u}) = I_{\bar{\mathcal{U}}_t}(\boldsymbol{u})$ if $X_t = 0$, the second line comes from Jensen's inequality for the absolute value function, and the last two lines come directly from the property of indicator functions. $\qquad\square$

## C.3  Proof of Proposition 2

*Proof.* In this part, we will show a stronger statement that, for all $\gamma > 0$,

$$\max_{\boldsymbol{u}:\|\boldsymbol{u}\|_2 \leq 1} -\text{Acc}(\boldsymbol{u}) + \frac{1}{T} \sum_{t=1}^{T} I_{\mathcal{U}_t(\gamma)}(\boldsymbol{u}) \leq 4\sqrt{\frac{\log(T)}{\underline{a}^2 \gamma^2 T}} + 6\sqrt{\frac{\log(T/\epsilon)}{T}},$$

where the uniform bound holds for all $\boldsymbol{u}$ in the unit ball (instead of unit sphere).

We will utilize Lemma 7 to prove the above inequality. For any $\boldsymbol{u}_0$ in the unit ball, let $\mathbb{Q} = \mathcal{N}(0, \tau^2 \boldsymbol{I}_n)$ and $\mathbb{Q}_0 = \mathcal{N}(\boldsymbol{u}_0, \tau^2 \boldsymbol{I}_n)$ be two normal distributions over the estimated parameter space, where $\boldsymbol{I}_n$ is the

$n$-dimensional identity matrix and $\tau$ is a constant to be determined. We remark that our choice of $\mathbb{Q}_0$ and $\mathbb{Q}$ will not affect the distribution of $(\boldsymbol{u}_t, \boldsymbol{a}_t, b)$, which depends on $\mathcal{P}_{\boldsymbol{a},b}$, $\mathcal{P}'_u$, $\delta$, and $\boldsymbol{u}^*$. Thus, by Lemma 7, the following inequality holds with probability no less than $1 - \epsilon$,

$$\mathbb{E}_{\mathbb{Q}_0}\left[\mathbb{E}\left[I_{(\mathcal{U}(\gamma))^c}(\boldsymbol{u})\right]\right] \leq \frac{1}{T}\mathbb{E}_{\mathbb{Q}_0}\left[\sum_{t=1}^{T} I_{(\mathcal{U}_t(\gamma))^c}(\boldsymbol{u})\right] + \sqrt{\frac{D_{KL}(\mathbb{Q}, \mathbb{Q}_0) + \log\frac{T}{\epsilon} + 2}{2T - 1}}, \qquad (25)$$

where $\mathcal{E}^c$ denotes the complement of a set $\mathcal{E}$. We note that on the left-hand-side, the inner expectation is taken with respect to the indicator function, and the outer expectation is taken with respect to $\boldsymbol{u} \sim \mathbb{Q}_0$.

Then we have

$$\mathbb{E}_{\mathbb{Q}_0}\left[\mathbb{E}\left[I_{(\mathcal{U}(\gamma))^c}(\boldsymbol{u})\right]\right] \leq \frac{1}{T}\mathbb{E}_{\mathbb{Q}_0}\left[\sum_{t=1}^{T} I_{(\mathcal{U}_t(\gamma))^c}(\boldsymbol{u})\right] + \sqrt{\frac{\frac{\|\boldsymbol{u}_0\|_2^2}{2\tau^2} + \log\frac{T}{\epsilon} + 2}{2T - 1}} \qquad (26)$$

$$\leq \frac{1}{T}\mathbb{E}_{\mathbb{Q}_0}\left[\sum_{t=1}^{T} I_{(\mathcal{U}_t(\gamma))^c}(\boldsymbol{u})\right] + 2\sqrt{\frac{\frac{\|\boldsymbol{u}_0\|_2^2}{2\tau^2} + \log\frac{T}{\epsilon}}{T}}.$$

Here, the first line is obtained by calculating of the KL-divergence between two Gaussian distributions, and the second line is a further simplification of the second line.

Now we analyze the left-hand-side and show that

$$I_{(\mathcal{U}(0))^c}(\boldsymbol{u}_0) - 2\exp\left(-\frac{a^2\gamma^2}{2\tau^2}\right) \leq \mathbb{E}_{\mathbb{Q}_0}\left[I_{(\mathcal{U}(\gamma))^c}(\boldsymbol{u})\right] \leq I_{(\mathcal{U}(2\gamma))^c}(\boldsymbol{u}_0) + 2\exp\left(-\frac{a^2\gamma^2}{2\tau^2}\right). \qquad (27)$$

To see this, for a fixed $(\boldsymbol{x}^*, \boldsymbol{a}, b)$, if $I_{\mathcal{U}(0)^c}(\boldsymbol{u}_0) = 1$, at least one inequalities from (22) to (24) is violated. If (22) does not hold for $\boldsymbol{u}_0$ while $I_{(\mathcal{U}(\gamma))^c}(\boldsymbol{u}) = 0$, there exists at least one $i \in \{1, ..., n\}$ such that

$$\frac{(\boldsymbol{u}_0)_i}{a_i} \geq 0, \ \frac{u_i}{a_i} < -\gamma.$$

The corresponding probability is no less than $\mathbb{P}((\hat{\boldsymbol{u}} - \hat{\boldsymbol{u}}_0)_i \leq -\underline{a}\gamma)$, which is bounded by $\exp(-\frac{a^2\gamma^2}{2\tau^2})$. Following the same analysis for (23) and (24), we have with probability no more than $2\exp(-\frac{a^2\gamma^2}{2\tau^2})$,

$$I_{\mathcal{U}^c}(\boldsymbol{u}) = 0, \ \text{while } I_{(\mathcal{U}(\gamma))^c}(\boldsymbol{u}_0) = 1,$$

which gives the left part of (27). The right part follows the same analysis.

Thus, let $\tau^2 = \frac{a^2\gamma^2}{2\log T}$. From (27) and (26), we have, with probability no less than $1 - \epsilon$, the following inequalities hold simultaneously for all $\boldsymbol{u}_0$ in the unit ball,

$$\mathbb{E}[I_{\mathcal{U}^c}(\boldsymbol{u}_0)] \leq \frac{1}{T}\sum_{t=1}^{T} I_{(\mathcal{U}(2\gamma))^c}(\boldsymbol{u}_0) + 4\exp(-\frac{a^2\gamma^2}{2\tau^2}) + 2\sqrt{\frac{\frac{\|\boldsymbol{u}_0\|_2^2}{2\tau^2} + \log\frac{T}{\epsilon}}{T}}$$

$$\leq \frac{1}{T}\sum_{t=1}^{T} I_{(\mathcal{U}(2\gamma))^c}(\boldsymbol{u}_0) + \frac{4}{T} + 2\sqrt{\frac{\log T}{\underline{a}^2\gamma^2 T} + \frac{\log(T/\epsilon)}{T}} \qquad (28)$$

$$\leq \frac{1}{T}\sum_{t=1}^{T} I_{(\mathcal{U}(2\gamma))^c}(\boldsymbol{u}_0) + 2\sqrt{\frac{\log T}{\underline{a}^2\gamma^2 T}} + 6\sqrt{\frac{\log(T/\epsilon)}{T}},$$

where the first inequality is obtained by plugging (27) into both sides of (26), the second line is obtained by plugging the value of $\tau$ into the inequality, and the last line is obtained by the convexity of the square

root function. Finally, with probability no less than $1 - \epsilon$

$$\sup_{\{\boldsymbol{u}:\|\boldsymbol{u}\|_2 \leq 1\}} -\mathrm{Acc}(\boldsymbol{u}) + \frac{1}{T}\sum_{t=1}^{T} I_{\mathcal{U}_t(\gamma)}(\boldsymbol{u}) = \sup_{\{\boldsymbol{u}:\|\boldsymbol{u}\|_2 \leq 1\}} \mathbb{E}[I_{\mathcal{U}^c}(\boldsymbol{u})] - \frac{1}{T}\sum_{t=1}^{T} I_{(\mathcal{U}_t(\gamma))^c}(\boldsymbol{u})$$

$$\leq 4\sqrt{\frac{\log T}{\underline{a}^2\gamma^2 T}} + 6\sqrt{\frac{\log(T/\epsilon)}{T}},$$

where the first line comes from the fact that $I_{\mathcal{A}}(\boldsymbol{u}) = 1 - I_{\mathcal{A}}(\boldsymbol{u})$ holds for any point $\boldsymbol{u}$ and set $\mathcal{A}$, and the second line comes from (28) with replacing $2\gamma$ by $\gamma$. $\qquad \square$

### C.4 Proof of Theorem 2

In this part, we assume that the optimal solution $\boldsymbol{x}_t^*$ satisfies

$$(\boldsymbol{x}_t^*)_i = 0 \text{ if } (\boldsymbol{u}_t)_i \leq 0 \text{ for all } t = 1, ..., T \text{ and } i = 1, ..., n.$$

This assumption is natural in that if purchasing the $i$-th item brings no positive utility, the customer will not purchase the item.

The proof is divided into two parts. In the first part, we show that, if $\gamma$ is sufficiently small, with high probability, the $\gamma$ margin indicator $I_{\mathcal{U}(\gamma)}(\boldsymbol{u}^*)$ is almost same as $I_{\mathcal{U}}(\boldsymbol{u}^*)$. In the second part, we combine the first part and Proposition 2 to draw the conclusion. Intuitively, the smaller $\gamma$ is, the smaller the difference between $I_{\mathcal{U}(\gamma)}(\boldsymbol{u}^*)$ and $I_{\mathcal{U}}(\boldsymbol{u}^*)$ we will have, and the worse the bound in Proposition 2 will be. Then the second part trades off between these two aspects and chooses the best $\gamma$. The formal proof is stated as below.

*Proof.* Let

$$\bar{\mathcal{U}} := \left\{ \boldsymbol{u} \in \mathcal{S}^{n-1} : \bar{\boldsymbol{x}}_t^* \text{ is an optimal solution of } \mathrm{LP}(\boldsymbol{u}, \boldsymbol{a}, b) \right\},$$

where $\bar{\boldsymbol{x}}_t^*$ is an optimal solution of $\mathrm{LP}(\boldsymbol{u}^*, \boldsymbol{a}, b)$. Now, let us show that $I_{\bar{\mathcal{U}}}(\boldsymbol{u}^*) = I_{\bar{\mathcal{U}}(\gamma)}(\boldsymbol{u}^*)$ holds with probability no less than $1 - \frac{4n^2\gamma}{\min\limits_{i:u_i^* \neq 0}|u_i^*|}$. By definition of $\bar{\mathcal{U}}$ and the assumption that $a_i \leq 1$ for all $i = 1, ..., n$, we have $\boldsymbol{x}^* \in \bar{\mathcal{U}}$ and

$$-\min_{\{i:\bar{x}_i^* > 0\}}\left(\frac{u_i^*}{a_i}\right) \leq -\min_{i:u_i^* \neq 0}|u_i^*|,$$

$$\max_{\{i:\bar{x}_i^* = 0, u_i^* \neq 0\}}\left(\frac{u_i^*}{a_i}\right) \leq -\min_{i:u_i^* \neq 0}|u_i^*|,$$

for any $\boldsymbol{a}$, $b$ and $\boldsymbol{u}^*$. Moreover, for any $i, j = 1, ..., n$ such that $u_i^*$, we have

$$\left|\frac{u_j^*}{a_j} - \frac{u_i^*}{a_i}\right| \leq \gamma \Leftrightarrow \frac{u_i^* a_j^*}{u_j^*} - \frac{a_i a_j}{u_j^*}\gamma \leq a_i \leq \frac{u_i^* a_j^*}{u_j^*} + \frac{a_i a_j}{u_j^*}\gamma,$$

which happens with probability no more than $\frac{4\bar{p}\gamma}{\min\limits_{i:u_i^* \neq 0}|u_i^*|}$. Since there are at most $n^2$ index pairs, with probability no less than $1 - \frac{4n^2\bar{p}\gamma}{\min\limits_{i:u_i^* \neq 0}|u_i^*|}$, we have

$$I_{\bar{\mathcal{U}}(\gamma)}(\boldsymbol{u}^*) = I_{\bar{\mathcal{U}}}(\boldsymbol{u}^*), \text{ for all } b, \gamma \leq \min_{i:u_i^* \neq 0}|u_i^*|, \text{ and fixed } \boldsymbol{u}^* \in \mathcal{S}^{n-1}. \qquad (29)$$

Then, similar to the proof of Lemma 1 with Hoeffding's inequality, we have

$$\mathbb{P}\left(\left|\sum_{t=1}^{T} I_{\bar{\mathcal{U}}_t(\gamma)}(\boldsymbol{u}^*) - \sum_{t=1}^{T} I_{\bar{\mathcal{U}}_t}(\boldsymbol{u}^*)\right| > 2n^2\bar{p}\gamma + \frac{\log T}{\sqrt{T}}\right) \leq 2\exp\left(-\frac{T\log T}{T}\right) = \frac{2}{T},$$

where the randomness in the above inequalities comes only from $(\boldsymbol{a}, b)$. Here, the inequality comes from the analysis of (29) and Hoeffding's inequality.

Next, we combine the above analysis and Proposition 2. Let $\hat{\boldsymbol{u}}$ be the optimal solution of the problem $OPT_\delta(\gamma)$. Then, we have, for any $\gamma \leq \min_{i:u_i^*\neq 0} |u_i^*|$, with probability no more than $1 - \epsilon - \frac{3}{T}$,

$$
\begin{aligned}
\text{Acc}(\hat{\boldsymbol{u}}) &\geq \sum_{t=1}^{T} I_{\mathcal{U}_t(\gamma)}(\hat{\boldsymbol{u}}) - 4\sqrt{\frac{\log T}{\underline{T}}} - 6\sqrt{\frac{\log(T/\epsilon)}{T}} \\
&\geq I_{\mathcal{U}_t(\gamma)}(\boldsymbol{u}^*) - 4\sqrt{\frac{\log T}{\underline{a}^2\gamma^2 T}} - 6\sqrt{\frac{\log(T/\epsilon)}{T}} \\
&\geq I_{\bar{\mathcal{U}}_t(\gamma)}(\boldsymbol{u}^*) - \delta - 4\sqrt{\frac{\log T}{\underline{a}^2\gamma^2 T}} - 7\sqrt{\frac{\log(T/\epsilon)}{T}} \\
&\geq I_{\bar{\mathcal{U}}_t}(\boldsymbol{u}^*) - \delta - \frac{4n^2\bar{p}\gamma}{\min_{i:u_i^*\neq 0} |u_i^*|} - 4\sqrt{\frac{\log T}{\underline{a}^2\gamma^2 T}} - 8\sqrt{\frac{\log(T/\epsilon)}{T}} \\
&\geq 1 - \delta - \frac{4n^2\bar{p}\gamma}{\min_{i:u_i^*\neq 0} |u_i^*|} - 4\sqrt{\frac{\log T}{\underline{a}^2\gamma^2 T}} - 8\sqrt{\frac{\log(T/\epsilon)}{T}},
\end{aligned}
\tag{30}
$$

where the first inequality is obtained by Proposition 2, the second line is obtained by the optimality of $\hat{\boldsymbol{u}}$, the third line is obtained by a similar statement as Proposition 1, the fourth line comes from (29), and the last line is obtained by the fact that $I_{\bar{\mathcal{U}}_t}(\boldsymbol{u}^*) = 1$ for all $t$. Finally, plug $\epsilon = \frac{1}{T}$ and $\gamma = \frac{1}{4nT^{1/4}}$ into (30). If $T$ is sufficient large such that $\frac{1}{4nT^{1/4}} \leq \min_{i:u_i^*\neq 0} |u_i^*|$, we have, with probability no less than $1 - \frac{4}{T}$,

$$\text{Acc}(\hat{\boldsymbol{u}}) \geq 1 - \delta - \frac{\bar{p}}{T^{1/4}} - 4\sqrt{\frac{n^4\log T}{\underline{a}^2 T}} - 16\sqrt{\frac{\log T}{T}} \geq 1 - \delta - \frac{n\bar{p}}{\min_{i:u_i^*\neq 0} |u_i^*| \cdot T^{1/4}} - 20\frac{n\log T}{\underline{a}T^{1/4}}.$$

$\square$

# D   Gaussian setting with Known Concentration Parameter

In this section, we revisit the Gaussian setting and discuss two methods to learn the mean vector $\boldsymbol{\mu}$ when the concentration parameter $\kappa$ is known. Specifically, we consider the following two cases : (i) We can design the constraint pair $(\boldsymbol{a}_t, b_t)$ for all $t = 1, ..., T$, (ii) We can only design $\boldsymbol{a}_t$ for all $t$ while we know a lower bound $\underline{b}$ such that $b_t \geq \underline{b}$ for all $t$. The basic idea is to estimate $\mathbb{P}(u_i > 0|\boldsymbol{\mu})$ for every $i = 1, ..., n$ and to use the idea of "moment matching" to identify $\boldsymbol{\mu}$.

**Constraint Design for the First Case**

For the first case, we set $\boldsymbol{a}_t = (1, ..., 1) \in \mathbb{R}^n$ and $b_t = n$ for all $t = 1, ..., T$. Then, at each time $t$, we have $(\boldsymbol{x}_t^*)_i = 1$ if and only if $(\boldsymbol{u}_t)_i > 0$. We can then estimate $\mathbb{P}(u_i > 0|\boldsymbol{\mu})$ by the sample average mean

$$\hat{p}_i := \frac{1}{T} \cdot \#\{t = 1, ..., T : (\boldsymbol{x}_t^*)_i = 1\}.$$

Next, we estimate $\boldsymbol{\mu}$ based on $\hat{p}_i$.

For any $i = 1, .., n$, let $\phi$ be an angle in $[0, \pi]$ satisfying $\cos \phi = \mu_i$. By Lemma 1 from Romanazzi (2014), we have

$$\mathbb{P}(u_i > 0 | \boldsymbol{\mu}) = \int_0^{\pi/2} \left( \frac{\kappa}{2\pi} \right)^{1/2} \frac{\sin^{(n-1)/2} \psi \cdot \tilde{I}_{(n-3)/2}(\kappa \sin \phi \sin \psi)}{\sin^{(n-3)/2} \phi \cdot \tilde{I}_{n/2-1}(\kappa)} \cdot \exp\left( \kappa \cos \phi \cos \psi \right) \mathrm{d}\psi, \qquad (31)$$

which depends only on $\mu_i$. Thus, with slight abuse of notation, we denote $\mathbb{P}(u_i > 0 | \mu_i)$ as the probability in (31). Moreover, if $n = 2$, we can have that the above function is strictly increasing. Then, by Lemma 1 from Romanazzi (2014) with the induction method, we can show that the above function is strictly increasing with respect to $\mu_i$ for every fixed $n$ and $\kappa$. For any fixed $n$ and $\kappa$, we first numerically compute (31). Then, a natural estimate of $\mu_i$ is

$$\hat{\mu}_i = \underset{\mu'_i \in [-1,1]}{\arg \min} |\mathbb{P}(u_i > 0 | \mu'_i) - \hat{p}_i|.$$

Then, $\hat{\boldsymbol{\mu}} = \{\hat{\mu}_1, ..., \hat{\mu}_n\}$ is our estimation of $\boldsymbol{\mu}$. We can further normalize $\hat{\boldsymbol{\mu}}$ in case that $\hat{\boldsymbol{\mu}} \notin \mathcal{S}^{n-1}$.

We remark that this method can be hardly generalized to the case that $\kappa$ is unknown. The reason is that we do not have a similar strictly increasing structure, and that the bijection between the probability (31) and the parameter $\boldsymbol{\theta}$ might not exist.

**Constraint Design for the Second Case**

Now we discuss how to design $\boldsymbol{a}$ if we cannot control $b$. One difficulty that prevents us applying the same method as in the previous section is that we might have $(\boldsymbol{x}_t^*)_i = 0$ while $(\boldsymbol{u}_t)_i > 0$ due to an insufficient budget. However, if we know a lower bound of $\{b_t\}_{t=1}^T$, we can still estimate $\mathbb{P}(u_i > 0 | \boldsymbol{\mu})$ by dismantling the high-dimensional estimation problem into a number of low-dimensional problems.

Denote the lower bound as $\underline{b}$. We set the first $\lceil \underline{b} \rceil$ entries in $\boldsymbol{a}$ be 1 and others be $\infty$, where $\lceil \cdot \rceil$ denotes the ceiling function. In this case, we have

$$(\boldsymbol{x}_t^*)_i > 0 \Leftrightarrow (\boldsymbol{u}_t)_i > 0 \text{ for all } i \leq \lceil \underline{b} \rceil \text{ and } t = 1, ..., T.$$

In this way, we can estimate $\mathbb{P}(u_i | \boldsymbol{\mu})$ for $i = 1, ..., \lceil \underline{b} \rceil$ following the previous case. To estimate the probability for other $i > \lceil \underline{b} \rceil$, we can divide the problem into $n/\underline{b}$ parts and estimate the probabilities separately.