# OpenReview forum: "Learning from Stochastically Revealed Preference"
_NeurIPS.cc/2022/Conference — NeurIPS 2022 Accept_

### Official Review · Reviewer_5mMn · 2022-06-30

**Rating:** 6
**Confidence:** 2
**Soundness:** 3 good
**Presentation:** 3 good
**Contribution:** 3 good

**Summary:**

This paper studies the problem of learning an known utility function via the actions (of an agent) in the stochastic setting, specifically a Gaussian setting and a $\delta$-corruption setting. Under both settings, methods with theoretical guarantees are proposed to learn the known utility function paramerized by a vector. Two respective algorithms are described and implemented for numerical evaluation to verify the theoretical guarantees (i.e., convergence).

**Questions:**

- Please see the first Weakness about "a set of agents".
- In Theorem 1, and in lines 130-132, why is it that $b$ no longer has a subcript $t$? What does $b$ (without the subscript $t$) refer to?
- In lines 137, "the Wasserstein distance in the theorem is not critical and it can be replaced with other distances such as the total variation distance and the Hellinger distance."
    - Does the bound in the theorem depend on the specific choice of distance in terms of _expression_ and _properties_ (e.g., does it need to be a proper metric)?
    - If multiple distances are possible, which should be used? In other words, what are some considerations one should make when picking the suitable distance?

**Limitations:**

The authors have identified in Sec. 5 that the approaches seem to scale poorly w.r.t. the dimension of the problem and argued that it can be mitigated by better and more efficient implementation. It would be good to briefly discuss this (perhaps in Appendix if there is no space) i) to pinpoint the component in the Algorithms that lead to this issue; ii) some intuitions for better implementation (e.g., why would parallel computing improve estimation accuracy, as I thought parallel computing is meant to distribute the computational load).

**Strengths And Weaknesses:**

This paper rigorously studies the problem of learning some unknown utility function under the stochastic setting (two specific settings). The problem setting is clearly laid-out and the proposed approaches seem sound (to the best of my ability). In particular, Theorems 1 and 2, provide the necessary theoretical guarantees for the approaches, respectively, which are later empirically verified. Regarding weaknesses, the setting of "a set of agents" can be made clearer and the motivation for considering the Gaussian setting and $\delta$-corruption setting togeher in one paper can be made clearer, too.

Strengths
- The overall theoretical rigor is good: in problem formulation, discussing the approaches and providing the necessary theoretical guarantees.
- The presentation is clear (i.e., writing and organization of content).

Weaknesses
- The setting of "a set of agents" is not so clear. A clear contrast (to existing works) this paper claims is "all the existing algorithms and analyses uder this topic relay on the assumption that all the agents share one common utility function (or a common utility parmeter vector) and thus can fail in the stochastic setting."
    - Are there multiple agents (with different utility functions) at the same time for the learner to learn? Why does stochasticity give rise to multiple utility functions (as opposed to one common utility function), since there is only one true $\mathcal{P}_{\mu}$ (for the Gaussian case) or $\boldsymbol{u}^*$ (for the $\delta$-corruption case)?
    - Following this, the motivation of this setting can be made clearer with references in addition to the described use-case in lines 28-30.
- The motivation for specifically considering both the Gaussian case and the $\delta$-corruption can be made clearer. Is it a standard approach to consider these two options (if so, are there references for it), or is there a significant motivation for considering these two case together in one paper?

---

> ### Author Response · Authors · 2022-08-02
> **Response to Reviewer 5mMn**
>
> We thank the reviewer for the comments. In the following, we address the comments one by one. We look forward to hearing your further feedback.
>
> For “a set of agents”, we are sorry for the previous confusion. “By sharing a common utility function”, we mean that the existing literature on preference learning and the general inverse optimization problem all consider the case that the objective/utility function is parametrized by an unknown but fixed deterministic vector $u=(u_1,…,u_n)$. Then, each agent/customer is represented by a sample tuple $(x_t^*,a_t,b_t)$ and all the agents share the same utility vector $u=(u_1,…,u_n)$. The objective is to identify this vector of $u$ from the available samples. Mathematically, under the linear program formulation, the problem of identifying $u$ is equivalent to finding a feasible solution to a system of linear inequalities. In comparison, we relax this assumption and allow some heterogeneity of the agents’ utilities. Specifically, each agent can have a different utility vector $u_t=(u_{t1},…,u_{tn})$, and the vector $u_t$ is sampled from some unknown distribution $P_u$. The goal then becomes to learn this distribution (instead of the previous deterministic vector) from samples.
>
> For the motivation of considering the Gaussian setting and the $\delta$-corruption setting, we made this choice mainly because of their analytical tractability. Modeling-wise, we also believe these two are the most natural candidates if one wants to relax the deterministic assumption. We refer to our response to Reviewer tzej for more detailed discussions on other alternatives and the connection between these two settings.
>
> “In Theorem 1, why is it that b no longer has a subscript t“.
> We are sorry for the typo, and there should be a $t$ for the subscript.
>
> “The dependence on the choice of distance”
> Yes. The bound depends on the choice of distance. Even for two equivalent metrics, the constant coefficients in the corresponding bounds can be different. Here, we remark that similar bounds in the same order with respect to the sample size also hold for the total variation distance, the Hellinger distance, and all other weaker metrics.
> The main required property of a metric to show our bound is the distinguishability of distributions in the corresponding metric space in terms of test functions. To be specific, we require a similar property in Lemma 9 to hold. Lemma 9 says that there exists a set of test functions distinguishing a given distribution and the true distribution at an exponential rate with respect to the sample size. Our bound depends on the rate of distinguishability.
>
> “Which distance to choose”
> If multiple distances are possible, the choice depends on our goal. For example, if we just hope to show a beautiful bound, we should consider a metric with better distinguishability, as discussed in the previous part; if we want to additionally recover the true parameter, we also need to consider the relationship between the metric of the distribution class and the metric of the parameter space.
>
> “The Limitation”
> Here, we will take the example of Algorithm 1 to address the question, and a similar reason is also applicable to Algorithm 2. The main reason why the algorithm performance is poor in the high dimensional setting is that we choose small numbers of iterations and sample sizes to allow the algorithm to finish within a limited time. If we run ten times more iterations with five times more samples, our new numerical result can increase to 75% for Experiment (ii) when n=25 in the Gaussian case. Thus, for question (ii), a better implementation can reduce the running time and allow us to run our algorithm with a larger number of iterations and a larger sample size within the same time, which will give a better result.
> One method to speed up our current method is to apply parallel computing in Step 5 to estimate the acceptance rate for the Monte Carlo method. That is why parallel computing can help improve the performance of Algorithm 1. For question (i), since the relationship between the implementation and the performance is discussed above, let us point out the most computational-consuming steps: Step 4, where the algorithm generates random numbers of the von Mises-Fisher distribution, and Step 5, where the algorithm estimates the acceptance rate by the Monte Carlo method.
>
> We look forward to hearing your further feedback and will get back timely in the following week.

---

> > ### Comment · Reviewer_5mMn · 2022-08-06
> > **Post-rebuttal**
> >
> > Thanks for the clarifications, most of my questions are addressed.
> >
> > I think it would be good to include the discussion on "which distance to select" the corresponding considerations in such selection; and a quantitative discussion on the complexity (i.e., which step of the algorithm is the most computationally costly and it depends on certain variables such as dimension, or the acceptance rate by the MC method in some way) if possible in the revision/appendix.
> >
> > Overall, I am maintaining my original rating.

---

### Official Review · Reviewer_tzej · 2022-07-03

**Rating:** 7
**Confidence:** 4
**Soundness:** 4 excellent
**Presentation:** 4 excellent
**Contribution:** 3 good

**Summary:**

This paper studies the problem of learning utilities from purchasing behavior, or, learning from revealed preference. While multiple papers have studied the sample complexity of backing out a single set of feasible item utilities that explain purchasing data drawn from a distribution, the authors study stochastic revealed preferences: the item utilities themselves are drawn from a distribution, and the dataset consists of purchases with respect to iid draws of the item utilities. The authors consider two main settings: in the first the utilities are drawn from a von Mises-Fisher distribution, in the second there is a deterministic utility vector that explains the purchase w.h.p, but the purchase (with complementary probability) might be explained by a utility vector drawn from an arbitrary distribution (representing a probability-delta corruption). They provide sampling and optimization techniques for both settings, with guarantees, and verify convergence of the learned utility parameters experimentally via actual implementations of the techniques.

**Questions:**

Is there any relation to the literature on learning stochastic choice? See, e.g., the manuscript “Prediction and Stochastic Choice” on SSRN and references therein. That literature deals mostly with binary preference relations so maybe there’s not much of a relation, but could be worth looking into (I’m not suggesting that these works be cited).

**Strengths And Weaknesses:**

I think this is an interesting contribution to the area of learning from choice data. The paper is nicely written and I was able to follow it with no issues. The techniques are interesting: the stochastic nature of the problem makes it more difficult than simply "backing out" a feasible utility vector. To this end I found the techniques used to overcome this challenge to be novel and interesting. I also appreciated the fact that the authors directly gave algorithms for arriving at the distribution P_u that could be experimentally verified, rather than stopping at sample complexity bounds.

One potential weakness is that the paper considers only two somewhat disparate settings, and there are not any connections (as far as I could tell) developed between the techniques used for both settings. It does seem like this setting is challenging and more general remarks would be tougher (e.g. there’s probably no general distribution-independent statement that can be made) but I wonder if there is a more general abstract learnability result in terms of parameters of P_u.

---

> ### Author Response · Authors · 2022-08-02
> **Response to Reviewer tzej**
>
> We thank the reviewer for the comments.
>
> For the comment on “two somewhat disparate settings,” we agree with the reviewers that the techniques we developed for the two settings do not have much in common. We chose these two settings mainly for modeling and analytical tractability considerations. In the following, we list a few other options that we have tried, and we will include these discussions in the future version of the paper:
>
> First, due to the scale-invariant property of the utility vector, another candidate is the compositional distribution that is supported on a standard simplex. This type of distribution is usually used for geoscience and environmental science to describe the decomposition of geochemical elements and soil contamination. The drawback of this distribution is that it requires the utility vector to be non-negative, and its learning tractability for our setting is not as good as the Gaussian distribution.
>
> Second, we also thought about the case that the utility vector takes a discrete distribution with finite support. When the support is known, the problem reduces to the learning/estimation of the probability parameters. A combination of probability parameters that is consistent with the observations can be learned through a simple maximum likelihood estimation procedure. When both the support and the probability parameters are unknown, this problem can be very challenging, and we will leave it for future studies.
>
> Third, there is a subtle connection between the two settings. The Gaussian setting can be viewed as a special case of the $\delta$-corruption setting when $(a_t,b_t)$’s are generated from certain distributions. Imagine that when the dispersion parameter $\kappa$ is large, the utility vectors sampled from the Gaussian distribution will concentrate around its mean direction. It can happen that all utility vectors within a certain neighborhood of this mean direction are not distinguishable from each other based on the observations of $(a_t,b_t)$’s. And all the other utility vectors outside this neighborhood can be viewed as a $\delta$ corruption. For a certain distribution of $(a_t,b_t)$, we can establish a relation between $\delta$, $\kappa$, and the diameter of this neighborhood. Then, the results from the $\delta$-corruption setting can also be applied to the Gaussian setting.
>
> We also thank the reviewer for pointing out the paper “Prediction and Stochastic Choice.” We were not aware of this work. As we understand, this paper is more aligned with the literature on choice modeling, which mainly focuses on the modeling of choosing one product among $m$ alternatives. From the perspective of choice modeling, our paper complements the existing works in providing a probabilistic model that allows purchasing multiple products.
>
> We look forward to hearing your further feedback and will get back timely in the following week.

---

### Official Review · Reviewer_qew4 · 2022-07-07

**Rating:** 5
**Confidence:** 2
**Soundness:** 3 good
**Presentation:** 2 fair
**Contribution:** 3 good

**Summary:**

This paper studies the problem of revealed preference under a stochastic setting. The utility of the agents follows an unknown distribution and we estimate the distribution from only the observation of the utility-maximizing actions of the agents. In this paper, the authors consider two underlying distributions: a Gaussian setting and a $\delta$-corruption setting, and provide Bayesian approaches for the problems with theoretical guarantees.

**Questions:**

- A typo in L261, where an extra "0" appears at the end of the sentence.
- Figure 1 plus Figure 2 appears to be too wide.

**Limitations:**

The authors seem to have addressed the limitations of their work.

**Strengths And Weaknesses:**

*Strengths:*

- The paper studies the revealed preference problem under a stochastic setting, where existing analyses assume that the agents share the same utility function.
- The authors propose Bayesian approaches for two utility function distributions and the method may potentially be extended to other settings beyond Gaussian and $\delta$-corruption settings.
- The authors also provide theoretical guarantees to the Bayesian models.

*Weaknesses:*

- The method requires sampling from a high-dimensional space, which incurs the curse of dimensionality that negatively impact the performance of the algorithms.

---

> ### Author Response · Authors · 2022-08-02
> **Response to Reviewer qew4**
>
> We thank the reviewer for the comments and for pointing out the typos. We will correct them in the later version of the paper. The high-dimensionality issue is also raised by another reviewer. We make the following remarks:
>
> First, for the Gaussian setting, we believe the difficulty in handling the high dimensional setting is intrinsic and might be inevitable for algorithm design. We can show that even for a low dimensional setting (n=2), if one uses the conventional maximum likelihood approach to estimate the parameter, the likelihood function is non-convex, and there exist locally optimal values that have an arbitrarily large gap compared to the global optimal. In this light, our Bayesian approach serves as a characterization for the posterior distribution: although one can’t say much about the landscape of the likelihood function, the posterior distribution will concentrate around the true parameter.
>
> Second, for the $\delta$-corruption setting, the high-dimensional issue can be resolved if we change our goal. Throughout the paper, our goal has been to identify the distribution of the utility vector. However, if we instead consider an online setting with the following objective:
> $\sum_{t=1}^T u^*_t x^*_t - u^*_t x_t,$
> where $x_t$ is the decision we made at time $t$ subject to the constraint $a_tx_t\le b_t, x_t\in [0,1]$. $x_t^*$ is the optimal solution of the linear program $\max u_t^*x_t,  s.t.  a_tx_t\le b_t, x_t\in [0,1]$. This objective characterizes a setting where the goal is to predict the customer’s choice and measure the prediction loss by the utility gap. Then the objective has a convexity structure, and the online gradient descent (GD) algorithm can be used, following the approach in (Bärmann et al., 2018, An Online-Learning Approach to Inverse Optimization). While the original paper of (Bärmann et al., 2018) studies a fixed utility setting, we can obtain a regret bound of $O(\sqrt{T}+\delta T)$ by analyzing the noisy version of the online GD algorithm.
>
> Thirdly, the problem can also be mitigated when the underlying problem has more structure, such as in Section D of the appendix, where we discuss the setting where the parameter $\kappa$ is known for the Gaussian setting. Also, when there exists one product that has a known deterministic utility, the problem may also be solved by learning all the other utility distributions one by one. This corresponds to the setting where this product with a known deterministic utility that represents the “no-purchase” option for the customer. We leave this type of problem as future research on what kind of additional structure enables efficient learning of the utility distribution, especially in a high-dimensional setting.
>
> We look forward to hearing your further feedback and will get back timely in the following week.

---

> > ### Comment · Reviewer_qew4 · 2022-08-07
> > **Post-rebuttal**
> >
> > Thank you for the clarification, and I will keep my rating for the paper.

---

### Official Review · Reviewer_9Dqv · 2022-07-10

**Rating:** 6
**Confidence:** 3
**Soundness:** 3 good
**Presentation:** 3 good
**Contribution:** 3 good

**Summary:**

The paper considers the stochastic setting where utility comes from some unknown distribution and the goal is to learn the distribution from the utility-maximizing actions of agents. They consider two distributions: Gaussian and delta-corruption. In both settings, they derive Bayesian approaches to learn distribution and provide some theoretical guarantees. They also provide approximation techniques and present some numerical experiments.

**Questions:**

I think paper can benefit from broader discussion on tractability and  inaccuracy in high dimension settings.

**Limitations:**

The authors address the limitations.

**Strengths And Weaknesses:**

Strenghts:
They motivate the guarantees based on posterior distribution using sound explanation and examples.

Weaknesses:
The experiments show that their approximation algorithms are quite weak and accuracy suffers in high dimensions.If we follow the authors argument that the primary reason is curse of dimensionality then this work only helps small dimensional settings.

---

> ### Author Response · Authors · 2022-08-02
> **Response to Reviewer 9Dqv**
>
> We thank the reviewer for the comments on the high dimensionality. We make the following remarks on the issue and will update the results in the later version of our paper.
>
> First, for the Gaussian setting, we believe the difficulty in handling the high dimensional setting is intrinsic and might be inevitable for algorithm design. We can show that even for a low dimensional setting (n=2), if one uses the conventional maximum likelihood approach to estimate the parameter, the likelihood function is non-convex, and there exist locally optimal values that have an arbitrarily large gap compared to the global optimal. In this light, our Bayesian approach serves as a characterization for the posterior distribution: although one can’t say much about the landscape of the likelihood function, the posterior distribution will concentrate around the true parameter.
>
> Second, for the $\delta$-corruption setting, the high-dimensional issue can be resolved if we change our goal. Throughout the paper, our goal has been to identify the distribution of the utility vector. However, if we instead consider an online setting with the following objective:
> $\sum_{t=1}^T u^*_t x^*_t - u^*_t x_t,$
> where $x_t$ is the decision we made at time $t$ subject to the constraint $a_tx_t\le b_t, x_t\in [0,1]$. $x_t^*$ is the optimal solution of the linear program $\max u_t^*x_t,  s.t.  a_tx_t\le b_t, x_t\in [0,1]$. This objective characterizes a setting where the goal is to predict the customer’s choice and measure the prediction loss by the utility gap. Then the objective has a convexity structure, and the online gradient descent (GD) algorithm can be used, following the approach in (Bärmann et al., 2018, An Online-Learning Approach to Inverse Optimization). While the original paper of (Bärmann et al., 2018) studies a fixed utility setting, we can obtain a regret bound of $O(\sqrt{T}+\delta T)$ by analyzing the noisy version of the online GD algorithm.
>
> Thirdly, the problem can also be mitigated when the underlying problem has more structure, such as in Section D of the appendix, where we discuss the setting where the parameter $\kappa$ is known for the Gaussian setting. Also, when there exists one product that has a known deterministic utility, the problem may also be solved by learning all the other utility distributions one by one. This corresponds to the setting where this product with a known deterministic utility that represents the “no-purchase” option for the customer. We leave this type of problem as future research on what kind of additional structure enables efficient learning of the utility distribution, especially in a high-dimensional setting.
>
> We look forward to hearing your further feedback and will get back timely in the following week.

---

### Meta-Review · Area_Chair_PDdv · 2022-08-24

**Recommendation:** Accept
**Confidence:** Certain

**Metareview:**

An interesting approach to stochastically revealed preferences

**Award:**

No

---

### Decision · Program_Chairs · 2022-09-14

Accept